# Plant-specific cochaperone SSR1 affects root elongation by modulating the mitochondrial iron-sulfur cluster assembly machinery

Xuanjun Feng [1,2]*, Yue Hu[2], Tao Xie[3], Huiling Han[4], Diana Bonea[5], Lijuan Zeng[6], Jie Liu [4], Wenhan Ying[3], Bona Mu[5], Yuanyuan Cai [3], Min Zhang[7], Yanli Lu[1], Rongmin Zhao [5]*, Xuejun Hua [3]*

**1** State Key Laboratory of Crop Gene Exploration and Utilization in Southwest China, Sichuan Agricultural University, Wenjiang, Sichuan, China, **2** Maize Research Institute of Sichuan Agricultural University, Wenjiang, Sichuan, China, **3** Key Laboratory of Plant Secondary Metabolism and Regulation of Zhejiang Province, College of Life Sciences and Medicine, Zhejiang Sci-Tech University, Hangzhou, Zhejiang, China, **4** Key Laboratory of Plant Resources, Institute of Botany, Chinese Academy of Sciences, Beijing, China, **5** Department of Biological Sciences, University of Toronto Scarborough, Toronto, Ontario, Canada, **6** Sichuan Keyuan Testing Center of Engineering Technology Co., Ltd; Chengdu, Sichuan, PR China, **7** Institute of Genetics and Developmental Biology, Beijing, China

* xuanjunfeng@sicau.edu.cn (XF); rongmin.zhao@utoronto.ca (RZ); xjhua@zstu.edu.cn (XH)

**Data Availability Statement:** All relevant data are within the paper and its Supporting Information files.

## Abstract

To elucidate the molecular function of *SHORT AND SWOLLEN ROOT1* (*SSR1*), we screened for suppressors of the *ssr1-2* (*sus*) was performed and identified over a dozen candidates with varying degrees of root growth restoration. Among these, the two most effective suppressors, *sus1* and *sus2*, resulted from G87D and T55M single amino acid substitutions in HSCA2 (At5g09590) and ISU1 (At4g22220), both crucial components of the mitochondrial iron-sulfur (Fe-S) cluster assembly machinery. SSR1 displayed a robust cochaperone-like activity and interacted with HSCA2 and ISU1, facilitating the binding of HSCA2 to ISU1. In comparison to the wild-type plants, *ssr1-2* mutants displayed increased iron accumulation in root tips and altered expression of genes responsive to iron deficiency. Additionally, the enzymatic activities of several iron-sulfur proteins and the mitochondrial membrane potential were reduced in *ssr1-2* mutants. Interestingly, *SSR1* appears to be exclusive to plant lineages and is induced by environmental stresses. Although HSCA2$^{G87D}$ and ISU1$^{T55M}$ can effectively compensate for the phenotypes associated with SSR1 deficiency under favorable conditions, their compensatory effects are significantly diminished under stress. Collectively, SSR1 represents a new and significant component of the mitochondrial Fe-S cluster assembly (ISC) machinery. It may also confer adaptive advantages on plant ISC machinery in response to environmental stress.

## Author summary

Iron-sulfur (Fe-S) clusters are crucial components found in many proteins that play essential roles in various biological processes. The machinery responsible for making these

**Funding:** This work was supported by the National Natural Science Foundation of China (32170307 to XH), the national Key R&D Program of China (2022YFD1201500 to YL), the Natural Science Foundation of Sichuan Province (2023NSFSC0221 to XF) and the NSERC Discovery Foundation (RGPIN-2019-07060 to RZ). The funders had no role in study design, data collection and analysis, decision to publish, or preparation of the manuscript.

**Competing interests:** The authors have declared that no competing interests exist.

clusters in mitochondria, known as ISC biosynthesis, has its origins in bacteria and remained largely unchanged through evolution. However, understanding of specific regulators that control Fe-S cluster production in plants is still limited. In this study, we identified a unique protein in plants, SSR1, which acts as a cochaperone. SSR1 facilitates the interaction between the two critical proteins, HSCA2 and ISU1, a necessary process for the release of Fe-S clusters from their scaffold. We also showed that SSR1 has evolved alongside the ISC biosynthetic machinery and that mutations in HSCA2 and ISU1 can compensate for its absence. This highlights the synergistic relationship between SSR1 and other components of the ISC machinery. Overall, this research uncovers a novel component of the ISC biosynthetic system and shows how it varies between plants, animals, and microorganisms.

## Introduction

Iron-sulfur (Fe-S) clusters are cofactors for many enzymes that play vital roles in biological processes like respiration, photosynthesis, DNA repair and hormone synthesis [1,2]. The cellular biosynthesis and assembly of Fe-S cluster is evolutionarily conserved from bacteria to higher plants and mammals [1,3]. Four Fe-S cluster biosynthesis pathways have been widely reported, namely ISC, CIA, SUF and NIF [4–8]. The eukaryotic Fe-S cluster assembly machineries (ISC in mitochondria and SUF in photosynthetic lineages) are of endosymbiotic origin from *Alphaproteobacteria* and *Cyanobacteria*, respectively [4–6]. The Eukarya-specific CIA system is specific for maturation of apoproteins located in both the cytosol and the nucleus [8]. The NIF machinery is specialized in the maturation of nitrogenase in N2-fixing bacteria [7]. Recently, two minimal Fe–S cluster assembly machineries, MIS (minimal iron–sulfur) and SMS (SUF-like minimal system), were reported as ancestral systems existing in *archaea* before the great oxidative event [9]. It has been proposed that in order to cope with Earth oxygenation and the increase in Fe-S client proteins, these ancestral systems underwent progressive complexification through the integration of new components, leading to the ISC, CIA, SUF, and NIF machineries [9].

The ISC machinery has been well-characterized in various species. The biogenesis of [2Fe-2S] clusters can be summarized into three main steps: first, the acquisition of sulfur from cysteine; second, the assembly and maturation of [2Fe-2S] clusters on scaffold proteins; and a third stage where [2Fe-2S] clusters are released from scaffold proteins and be inserted into an apo-proteins [4–8]. Desulfurases, such as IscS/Nfs1, catalyze the release of sulfur from L-cysteine and transfers it to the scaffold protein IscU/Isu1. In eukaryotes, ISD11 is reported to be essential for this process [8,10–12]. While frataxin/Yfh1/cyaY were previously thought to be iron donors, recent studies suggest they function more as accelerators of persulfide transfer rather than direct iron donors [3,5,6,8,13]. In addition, ferredoxin (such as fdx/Yah1) and its reductase (such as Arh1) are essential for the maturation of [2Fe-2S] clusters [3,8]. Once assembled, [2Fe-2S] clusters are released from IscU/Isu1 and transferred to downstream transfer protein, such as glutaredoxin Grx5, and recipient apo-proteins [4–8]. During this stage, the Hsp70–Hsp40 (DnaK–DnaJ) type chaperone/cochaperone system (such as HscA/Ssq1 and HscB/Jac1) are required [8,14,15]. HscB/Jac1 promotes HscA/Ssq1 to bind with the conserved hydrophobic LPPVK motif of IscU/Isu1 [16,17]. The cluster handover from IscU/Isu1 to Grx5 may be facilitated by their simultaneous binding to the HscA/Ssq1 chaperone on different interaction sites [18]. This process is energized by the hydrolysis of ATP by HscA/Ssq1, which is generally stimulated by the binding of HscB/Jac1 and/or IscU/Isu1 [19–22].

The Fe-S cluster assembly machineries are further complicated in higher plants than in microorganisms. Many Arabidopsis genes encoding the core components of ISC machinery have been identified based on sequence homology with their counterparts in bacteria or yeast, and some core proteins are encoded by a multiple-gene family [2,23–26]. For instance, there are three genes encoding cysteine desulfurase in Arabidopsis, namely *NFS1*, *NFS2* and *ABA3* [2,27]. Only NFS1 showed mitochondrial localization and high homology with IscS, a bacterial cysteine desulfurase [25]. Scaffold protein ISU is also encoded by three genes, *ISU1*, *ISU2* and *ISU3*, in Arabidopsis, with *ISU1* being expressed most abundantly [24,25]. Moreover, like those in bacteria, Arabidopsis HSCA1, HSCA2, HSCB and ISU1 could interact with each other, and the binding with HSCB and/or ISU1 also promotes the ATPase activity of HSCA2 [22]. Two DnaJ proteins, DJA6 and DJA5, have recently been identified as essential for chloroplast iron–sulfur cluster biogenesis by facilitating iron incorporating. Notably, DJA6 and DJA5 are unique to photosynthetic organisms, ranging from cyanobacteria to higher plants [28].

The complexity of iron-sulfur cluster synthesis in higher plants reflects the importance of these cofactors in plant biology and the necessity for precise regulation. Despite the identification and extensive study of plant ISC genes, the question of whether plants possess any specific component for precise regulation of the ISC machinery remains unanswered. Previously, we characterized an Arabidopsis gene called *SHORT AND SWOLLEN ROOT1* (*SSR1*), which encodes a mitochondrial protein containing a tetratricopeptide repeat (TPR) domain. This gene has been shown to regulate root growth [29] and mitochondrial function [30]. The *ssr1-2* mutant exhibited pleiotropic growth phenotypes, including short roots, brush-like roots, bushy shoots, and growth retardation, along with altered expression patterns of various auxin transport or response-related proteins [29]. Here, by screening and analyzing the suppressors of *ssr1-2*, we provided evidence that SSR1, acting as a cochaperone-like component in the Arabidopsis ISC machinery, interacts with both ISU1 and HSCA2 to promote their interaction. In addition, *SSR1* is induced by environmental stress stimuli and is uniquely present in plant lineages. Its homolog could not be found in organisms of other kingdoms. Therefore, we propose that *SSR1* may be a plant-specific molecular cochaperone that critically regulates the ISC machinery and confers adaptive advantages in response to environmental stress.

## Results

### *sus1* and *sus2* are mutations that change one amino acid in HSCA2 and ISU1

To better understand the molecular mechanism of *SSR1* in affecting root development, the loss-of-function mutant *ssr1-2* (FLAG_356A08) was mutagenized by ethyl methanesulfonate and the M2 progenies were used for screening suppressors of the short-root phenotype of *ssr1-2*. From over a dozen suppressors, two lines, designated as *ssr1-2 sus1* and *ssr1-2 sus2* (*sus*: <u>sup-pressor of ssr1-2</u>), have the best recovered root length (about 85% and 100%, respectively) at seedling stage as well as a complete rescue of the dwarf phenotype at the flowering stage (Fig 1A and 1D).

It has been previously reported that several auxin-related markers were abnormally expressed in *ssr1-2* seedlings [29]. To examine whether *sus1* and *sus2* also rescued the abnormal expression of some auxin-related genes, *PIN1-GFP*, *PIN2-GFP*, *Wox5-GFP* and *DR5-GFP* marker lines in *ssr1-2* background were crossed to double mutants *ssr1-2 sus1* and *ssr1-2 sus2*. The resulting F3 progenies which are homozygous for both *sus* and marker genes were identified. Confocal microscopic analyses showed that the expression patterns of all the four auxin-related genes in double mutants were restored and resembled those observed in the wild-type

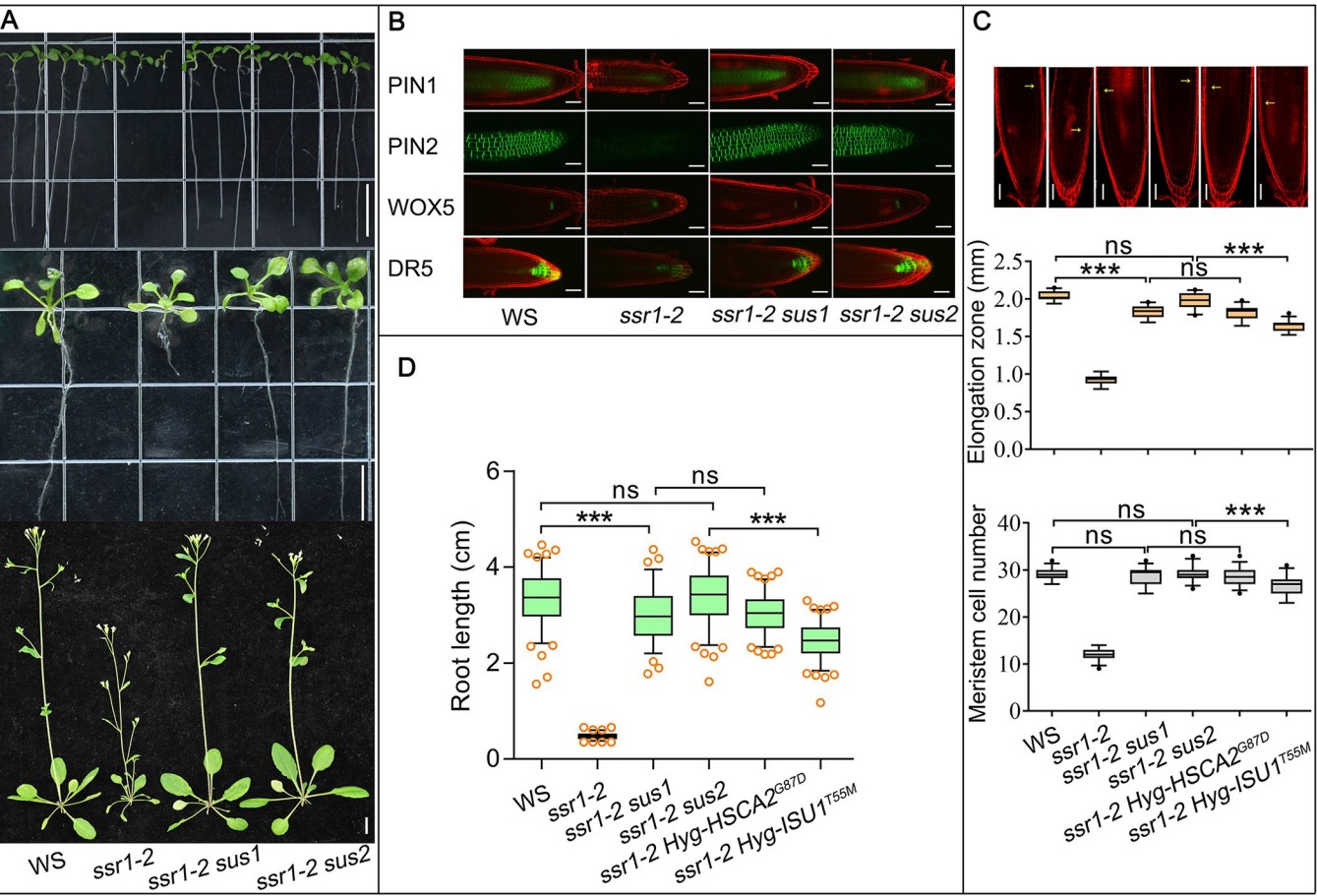

**Fig 1. Identification of *ssr1-2* suppressor mutant genes *sus1* (*HSCA2^{G87D}*) and *sus2* (*ISU1^{T55M}*).** Two suppressors (*sus1* and *sus2*) were identified to rescue the growth defect of *ssr1-2* and the whole genome resequencing identified *sus1* as *HSCA2^{G87D}* while *sus2* as *ISU1^{T55M}*. (A) *sus1* and *sus2* rescued both the root and shoot growth defects of *ssr1-2* to the levels comparable to wild type (*WS: Wassilewskija*). Top, 10-day-old seedlings; middle, 20-day-old seedlings; bottom, 40-day-old plants. Scale bar = 1 cm. (B) *sus1* and *sus2* rescued the expression defects in *ssr1-2* of auxin-related marker genes *PIN1::PIN1GFP*, *PIN2::PIN2GFP*, *WOX5::WOX5GFP*, and *DR5::GFP*. These genes were labelled as PIN1, PIN2, WOX5, and DR5, respectively, and the levels of expression were found to be comparable to those observed in WS. (C) *sus1* and *sus2* mutant genes were cloned, designated as *HSCA2^{G87D}* and *ISU1^{T55M}*, respectively, and re-transformed back to *ssr1-2* plants to rescue the short root phenotype. Top, images of meristem zone of different lines at 10-day-old. Middle, elongation zone length, bottom, meristem zone cell number. In transgenic lines, the transgenes are associated with a hygromycin resistant gene and therefore designated as *Hyg-HSCA2^{G87D}* or *Hyg-ISU1^{T55M}*. It should be noted that *ssr1-2 Hyg-HSCA2^{G87D}* still contains the wild type *HSCA2* allele and *ssr1-2 Hyg-ISU1^{T55M}* contains wild type *ISU1* allele. (D) Primary root length of different lines grown for 10 days. Lines analyzed are the same as in 1*C*. Scale bars in 1*B* and 1*C* are 50 μm. Yellow arrows point to the end of meristem zone in top images of 1*C*. In 1*C* and 1*D*, 15 and 100 roots of each were used for statistical analysis by ANOVA. Statistical value is shown as box & whiskers with 5–95 percentile. Asterisk "*", "***" and "****" represent statistics analysis with *p* values < 0.05, 0.01, 0.001 respectively. "ns" represents statistically not significant.

background (Fig 1B). Correspondingly, the defect of *ssr1-2* in meristem cell number and elongation zone observed in the *ssr1-2* mutant was also restored by *sus1* and *sus2* (Fig 1C). These data indicate that *sus1* and *sus2* are two suppressor mutants that could indeed rescue the defects of *ssr1-2* and therefore were chosen for further analysis.

To investigate the genetic nature of *sus1* and *sus2*, we conducted a cross-breeding in which *ssr1-2 sus1* and *ssr1-2 sus2* were crossed with *ssr1-2*, respectively. All *ssr1-2 sus1/SUS1* F1 seedlings exhibited similar root length to *ssr1-2 sus1* suggesting that *sus1* may be a dominant mutation (S1A Fig). In contrast, the primary roots of *ssr1-2 sus2/SUS2* F1 seedlings were shorter than that of *ssr1-2 sus2*, indicating that *sus2* may be a semi-dominant mutation (S1A Fig). Further analysis of self-pollinated F2 progenies of *ssr1-2 sus1/SUS1* and *ssr1-2 sus2/SUS2*

confirmed our speculations, with clear 3:1 and 1:2:1 segregation, respectively, in terms of the root lengths (S1B and S1C Fig and S1 Table).

A total of five candidate genes were selected from each suppressor line based on super bulked segregant analysis (S2 and S3 Tables). These genes were then verified by genetic complementation. Since *sus1* and *sus2* are dominant and semi-dominant mutations, respectively, the complementation analysis was performed by re-introducing the full-length genomic fragments of candidate genes from the suppressor lines into *ssr1-2*. It turned out that the reintroduction of *At5g09590* from *ssr1-2 sus1* and *At4g22220* from *ssr1-2 sus2* was the only means of rescuing the short root phenotype and the defects of apical meristem cell number and elongation zone observed in *ssr1-2* (Fig 1C and 1D), thus confirming that *At5g09590* and *At4g22220* are *SUS1* and *SUS2*, respectively. *At5g09590* and *At4g22220* encode a mitochondrial heat shock cognate 70 (mtHSC70-2 or HSCA2) and an ISU1 protein, respectively. Both proteins are core components of the ISC machinery involved in the assembly of Fe-S clusters in mitochondria [2,22]. *sus1* and *sus2* each bears a point substitution mutation in the coding region of the corresponding *HSCA2* and *ISU1* genes. *sus1* carries a G to A transition at position 260, resulting in an HSCA2$^{G87D}$ mutant protein. *sus2* carries a C to T transition at position 164, resulting in an ISU1$^{T55M}$ mutant protein.

With the identification of the two suppressor genes which cause amino acid substitution in HSCA2 (HSCA2$^{G87D}$) and ISU1 (ISU1$^{T55M}$), we took a different approach to analyze the other candidate suppressors obtained from our initial screen. *HSCA2* and *ISU1* genes from other suppressor lines were all cloned and re-sequenced. Interestingly, only *sus3* does not carry mutations in *HSCA2* or *ISU1*. The additional suppressors carry single point mutations either in *HSCA2* or *ISU1*, resulting in ISU1$^{A143V}$, ISU1$^{G106D}$, ISU1$^{A143T}$, HSCA2$^{R394C}$ and ISU1$^{A140V}$, respectively (S4 Table). Similar to *sus1* and *sus2*, the suppressor function of ISU1$^{G106D}$ and ISU1$^{A143T}$ were investigated and verified by genetic complementation (S2A Fig).

HSCA1 is a homolog of HSCA2, and also located in mitochondria [2]. The *hsca1* mutant was recently reported to have a deficiency in embryo development and root growth as well [31,32]. To test whether HCSA1 performs a similar function to HSCA2 in root growth, a mutant form of HSCA1, designated HSCA1$^{G82D}$, was created and introduced into *ssr1-2* mutants under the control of its native promoter. This mutant corresponds to HSCA2$^{G87D}$. The short-root phenotype of *ssr1-2* seedlings was substantially suppressed by the expression of HSCA1$^{G82D}$ (S2B Fig), suggesting that HSCA1 may have overlapping function with that of HSCA2 *in vivo*. Furthermore, we obtained a knock-out mutant of *HSCA2*, *hsca2* (CS479451 from ABRC), and observed that *hsca2* seedlings did not exhibit any significant phenotype regarding primary root length (S2C Fig). All these data indicate HSCA1 and HSCA2 are functionally redundant in root development, and HSCA1 likely plays a dominant role.

We would like to understand whether *sus1* or *sus2* can promote root growth in the wild-type background. To achieve this, we isolated single mutants of *sus1* and *sus2* after crossing *ssr1-2 sus1* and *ssr1-2 sus2* with the wild-type *Wassilewskija* (WS). Then the root length of these mutants was investigated. Under the optimal growth conditions, the root length of the *sus1* and *sus2* single mutants was comparable to that of the wild type (S2D Fig), implying that the identified suppressors specifically rescued the defect caused by the *SSR1* mutation rather than promoting root growth in a parallel pathway.

## SSR1 interacts with HSCA2 and ISU1 and promotes HSCA2-ISU1 association

Since both suppressor genes, *HSCA2* and *ISU1*, encode components in ISC machinery, it is possible that SSR1 may be also involved in Fe-S cluster assembly in mitochondria. To

understand the function of *SSR1* in the ISC machinery, we analyzed possible protein-protein interactions between SSR1, HSCA2 and ISU1 by bimolecular fluorescence complementation (BiFC) assay using Arabidopsis mesophyll protoplasts. Fluorescence signals were clearly visible in protoplasts as discrete bright spots that overlapped with mitochondria when cCFP-tagged SSR1 was co-expressed with nVenus-tagged HSCA2 or ISU1 (Fig 2A).

Co-IP assays were then performed using transgenic plants to verify above interaction *in vivo*. Given the relatively weak activity of the native *SSR1* promoter, the *SSR1-Flag* construct driven by the 35S promoter was employed and co-integrated into *ssr1-2* mutants with the *HSCA2-Myc* construct driven by the native promoter via transformation and subsequent crossing. Protein extracts from transgenic plants were incubated with anti-Flag antibody resin to coprecipitate SSR1-Flag associated proteins. Subsequent immunoblotting analysis with anti-Myc or anti-ISU1 antibodies demonstrated that both HSCA2-Myc and ISU1 were coprecipitated with SSR1-Flag, though with much less ISU1 being co-purified compared with HSCA2 (Fig 2B).

Additionally, protein-protein interaction was further investigated by *in vitro* pull-down assays using purified proteins. A variety of combinations of the proteins His-HSCA2-Myc, His-HSCA2$^{G87D}$-Myc, His-SSR-Myc, SSR-Myc, ISU1-Myc, and ISU1$^{T55M}$-Myc were incubated, followed by pull-down with Ni-Sepharose and analyzed by Western blot with anti-Myc antibody. The results further demonstrated that HSCA2, SSR1, and ISU1 can interact directly with one another (Fig 2C). It is noteworthy that ISU1 exhibited a more robust interaction with HSCA2$^{G87D}$ than with HSCA2, while HSCA2 demonstrated a stronger interaction with ISU1$^{T55M}$ than with ISU1 (Fig 2C). Additionally, the presence of SSR1 further promoted the interaction between ISU/ISU1$^{T55M}$ and HSCA2/HSCA2$^{G87D}$ (Fig 2C).

The ISC machinery is a dynamic and complex system composed of multiple components. We postulate that the effect of SSR1 on the affinities of protein interaction between ISU/ISU1$^{T55M}$ and HSCA2/HSCA2$^{G87D}$ may be more pronounced in the presence of other cellular cofactors. Therefore, pull-down assay was also tried with purified His-HSCA2-Myc/His-HSCA2$^{G87D}$-Myc proteins and *E. coli* lysates containing SSR-Myc, ISU1-Myc, and ISU1$^{T55M}$-Myc, respectively. As expected, ISU1 exhibited a more robust interaction with HSCA2$^{G87D}$ than with HSCA2, while HSCA2 demonstrated a stronger interaction with ISU1$^{T55M}$ than with ISU1, and the presence of SSR1 significantly promoted the interaction between ISU/ISU1$^{T55M}$ and HSCA2/HSCA2$^{G87D}$ (S3 Fig).

We tried to model the interaction between HSCA2 and ISU1 using AlphaFold3 to understand the effect of mutations on the interactions. Notably, previous studies have identified the LPPVK sequence in ISU1 as a critical site for HSCA2 binding [16,17,19]; however, this LPPVK region does not resided at the interaction interface with HSCA2 in the predicted interaction model, underscoring the misalignment in the predicted relative orientations of the two proteins (S1 Dataset). Therefore, it is not feasible to compare the effects of amino acid variations on the interaction between ISU1 and HSCA2 using this method.

Next, the combination of blue native polyacrylamide gel electrophoresis (BN-PAGE) and SDS-PAGE was employed to investigate the potential for His-SSR1-Myc, His-HSCA2-Myc, and ISU1-Myc proteins to form complexes. Although the immunoblotting signal is weak, the signal of three proteins is clearly observed at approximately 400 kDa simultaneously (S4A Fig). However, the co-existence of His-SSR1-Myc, His-HSCA2-Myc, and ISU1-Myc at approximately 400 kDa was not observed in any other combinations (S4B–S4G Fig). These results indicate that the HSCA2-SSR1-ISU1 complex does exist, but it may be not stable.

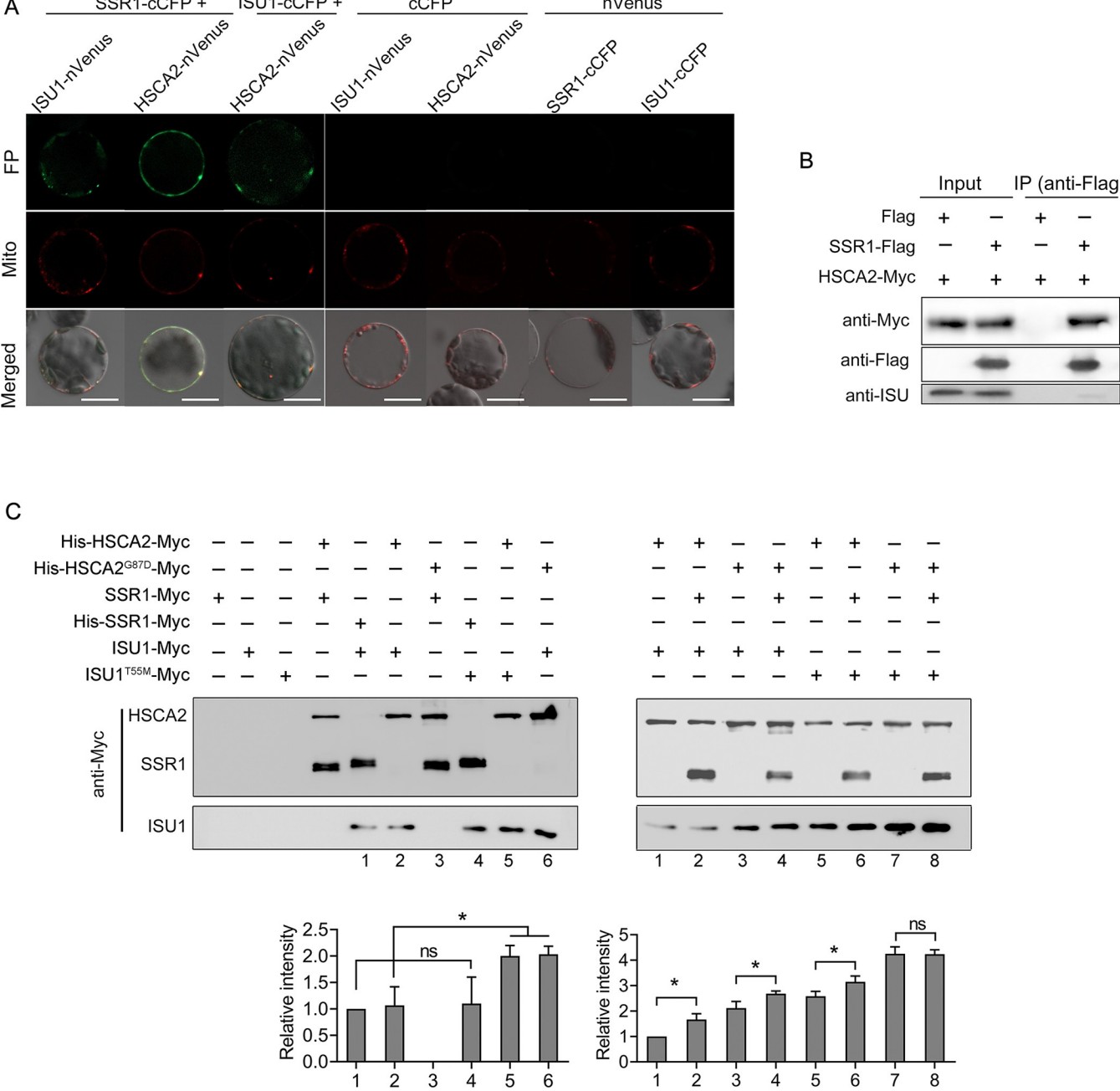

**Fig 2. SSR1 interacts with HSCA2 and ISU1 and promotes HSCA2-ISU1 association.** (*A*, *B*) *in vivo* interactions between SSR1 and HSCA2 or ISU1 were analyzed in Arabidopsis protoplasts by BiFC and co-immunoprecipitation assays. In (*A*), full length of SSR1, HSCA2 or ISU1 was used for fusion with reporter genes. The mitochondria are labeled by MitoTracker and emit red light. FP represents reconstitute fluorescent proteins. In (*B*), stable transgenic lines co-expressing Flag-tagged full-length SSR1 or Flag alone under CaMV 35S promoter and Myc-tagged full-length HSCA2 under its native promoter were immunoprecipitated with anti-Flag antibody, and the co-purified proteins were immunodetected with anti-Myc and anti-ISU antibodies. (*C*) In vitro pull-down assays were conducted using Ni-NTA Sepharose with His-tagged proteins expressed and purified from *E. coli* as baits. The prey proteins were also expressed and purified from *E. coli*, after which the His-tag was excised. Co-purified proteins were detected with anti-Myc antibody. The immunoblotting signals of ISU1-Myc or ISU1$^{T55M}$-Myc were quantified using the ImageJ software, and the relative intensities are presented under the respective lanes with three replicates. The mitochondrial targeting peptide coding sequences of HSCA2, SSR1 and ISU1 were removed for protein expression in *E. coli*. The statistical significance of the results was determined through the use of a student *t*-test. Variations were considered significant if P <0.05(*), 0.01(**) or 0.001(***).

## The SSR1 function is dependent on the interaction between HSCA and ISU1

Based on the protein-protein interaction, we speculated that the enhanced affinity between ISU1 and HSCA2, when a point mutation in either ISU1 (ISU1$^{T55M}$) or HSCA2 (HSCA2$^{G87D}$) is present, may be responsible for the suppression of the root growth defect in *ssr1-2*. To verify this hypothesis, amino acid substitutions were introduced into ISU1$^{T55M}$ at the LPPVK motif, which was previously reported to mediate the interaction with HSCA [16,17,19]. In vitro pull-down assays showed that L126A, P127A, or P128S mutation slightly impaired the interaction between ISU1$^{T55M}$ and HSCA2, while V129E or K130A mutation dramatically reduced the affinity between ISU1$^{T55M}$ and HSCA2 (Fig 3A). In addition, simultaneous substitution of PVK to AAA almost completely abolished the interaction between ISU1$^{T55M}$ and HSCA2 (Fig 3A). Subsequently, the ISU1 mutant constructs were introduced into *ssr1-2* plant to test their ability to suppress the short-root phenotype of *ssr1-2*. It is interesting to note that ISU1$^{T55ML126A}$, ISU1$^{T55MP127A}$, and ISU1$^{T55MP128S}$, demonstrate partial rescue of *ssr1-2*, whereas ISU1$^{T55MV129E}$, ISU1$^{T55MK130A}$, and ISU1$^{T55MAAA}$ are unable to do so (Fig 3B). This aligns with their capacity to interact with HSCA2. Taken all these together, we have demonstrated that enhanced interaction between ISU1 and HSCA2 is likely the cause of the rescue of *ssr1-2* phenotype.

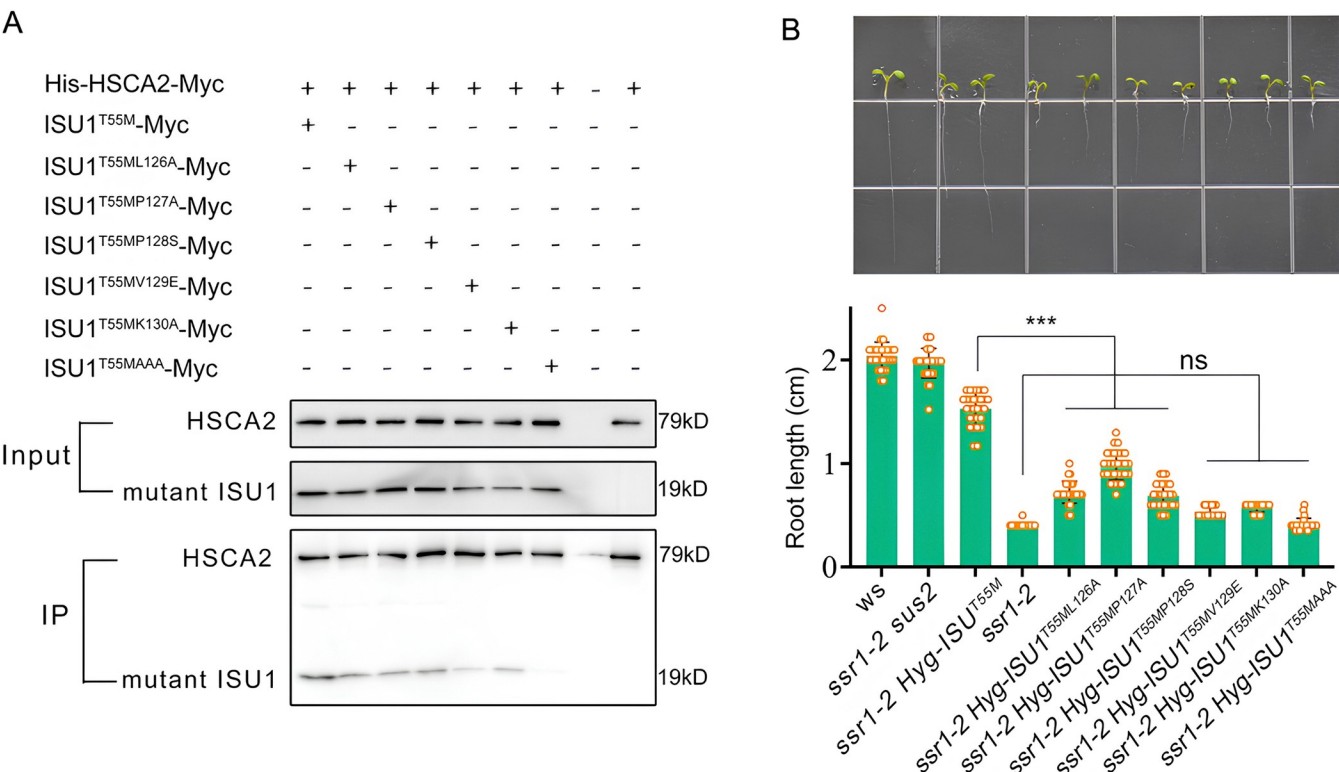

**Fig 3. Interaction between ISU1$^{T55M}$ and HSCA2 is essential for the rescue of *ssr1-2* growth defect by ISU1$^{T55M}$.** (*A*) Ni-NTA in vitro pull-down assays with His-tagged HSCA2 expressing in *E. coli* as the bait. Different Myc-tagged ISU1 mutation variants within the LPPVK motif were expressed in *E. coli* and differentially co-purified with HSCA2. The mitochondrial targeting peptide coding sequences of HSCA2 and ISU1 were removed for protein expression in *E. coli*. (*B*) Root length of the wild type (WS) and *ssr1-2* mutants carrying *sus2* allele or transformed with an additional *ISU1$^{T55M}$* gene with differentially mutated LPPVK motif. Here, the full length of ISU1 with different mutations was employed for transformation. Top, representative seedlings at 8-days-old. Bottom, root lengths shown as bar graphs with error bars representing standard deviations from about 30 seedlings. Statistical significance was determined through student *t*-test. The asterisk "***" represents a statistically significant result with *p*-values < 0.001. The symbol "ns" represents a statistical insignificant result with *p*-values < 0.05.

## SSR1 displays a chaperone-like function

The involvement of HSCA and HSCB chaperone system in the biosynthesis of Fe-S clusters has been well documented in bacteria, yeast and plant [2,7,14,22]. The release of mature Fe-S cluster from ISU1 requires the binding of chaperone HSCA, and this process can be stimulated by cochaperone HSCB [14]. Since we have shown that SSR1 interacts with HSCA2 and ISU1, and promotes the interaction between ISU1 and HSCA2, we speculate that SSR1 may display a chaperone-like activity. To explore this possibility, we used purified SSR1, ISU1, HSCA2 and HSCA2$^{G87D}$ from *E. coli* to test their general chaperone activity in preventing heat-induced substrate protein from aggregation.

By using citrate synthase (CS) as a model substrate, it was shown that both HSCA2 and HSCA2$^{G87D}$ have a strong general chaperone activity, and the difference between the wild-type and the mutant form is subtle (Figs 4A and S5). ISU1, however, unexpectedly inhibited heat-induced aggregation of CS, although not as efficiently as HSCA2 (Fig 4B). When both ISU1 and HSCA2/HSCA2$^{G87D}$ were added to CS, much more aggregates accumulated especially at the later time (Fig 4B and 4C). This implied that certain physical interactions occurred between HSCA2 and ISU1. Interestingly, whenever SSR1 is present, no aggregate formed, indicating SSR1 has a powerful chaperone-like activity (Fig 4B and 4C). Further titration analysis demonstrated that SSR1 itself displayed a very strong chaperone activity in preventing heat-induced citrate synthase (CS) from aggregating even at a very low concentration (Fig 4D).

## SSR1 affects iron homeostasis

The ISC machinery plays a key role in maintaining cellular iron homeostasis by regulating the uptake, storage, and distribution of iron [33,34]. This helps to prevent iron overload or deficiency, which can lead to oxidative stress and damage to cellular structures. Overexpression of *AtHSCB* led to higher iron accumulation in roots and lower iron content in shoots, whereas *hscb* knockdown mutants exhibited reduced iron uptake in roots but increased iron levels in shoots [35]. Perls' prussian blue staining showed that *ssr1-2* has an unusually high iron content in the root tips (Fig 5A). Iron content in root tips and leaves was further measured using inductively coupled plasma-mass spectrometry (ICP-MS). *ssr1-2* had higher iron content in the tips and leaves than the wild type (Fig 5B and S5 Table). Furthermore, the accumulation of iron around the quiescent center and the stem cell niche of the root was obviously higher in *ssr1-2* than in the wild-type (Fig 5C). As expected, *SSR1*, *sus1*, and *sus2* well restored the iron accumulation phenotype in the root tips and leaves of *ssr1-2* mutants (Fig 5A–5C and S5 Table).

## Loss-of-function of *SSR1* causes mitochondrial dysfunction

The interaction of HSCA and ISU are important for the release of Fe-S clusters from ISU to apo-proteins [2,20]. Additionally, our recent study on another weak allele of *SSR1* indicated that *SSR1* is important in regulating the function of mitochondrial electron-transport chain complexes, a few of which are well-known Fe-S proteins [30]. We therefore hypothesized that the mitochondrial Fe-S cluster biosynthesis or the biosynthesis of Fe-S proteins are impaired in the *ssr1-2* mutant.

We then investigated the activities and/or protein levels of some Fe-S proteins, namely mitochondrial complex I (CI), aconitase (ACO) and succinate dehydrogenase 2 (SDH2), in *ssr1-2* seedlings. Given that aconitase is located in mitochondria and cytosol, the aconitase activity from both compartments was measured. The results showed that the enzymatic activities of CI, aconitase, and the protein level of SDH2 in *ssr1-2* were dramatically low compared to that in the wild type, *ssr1-2 sus1*, and *ssr1-2 sus2* (Fig 6A–6C). However, the enzymatic

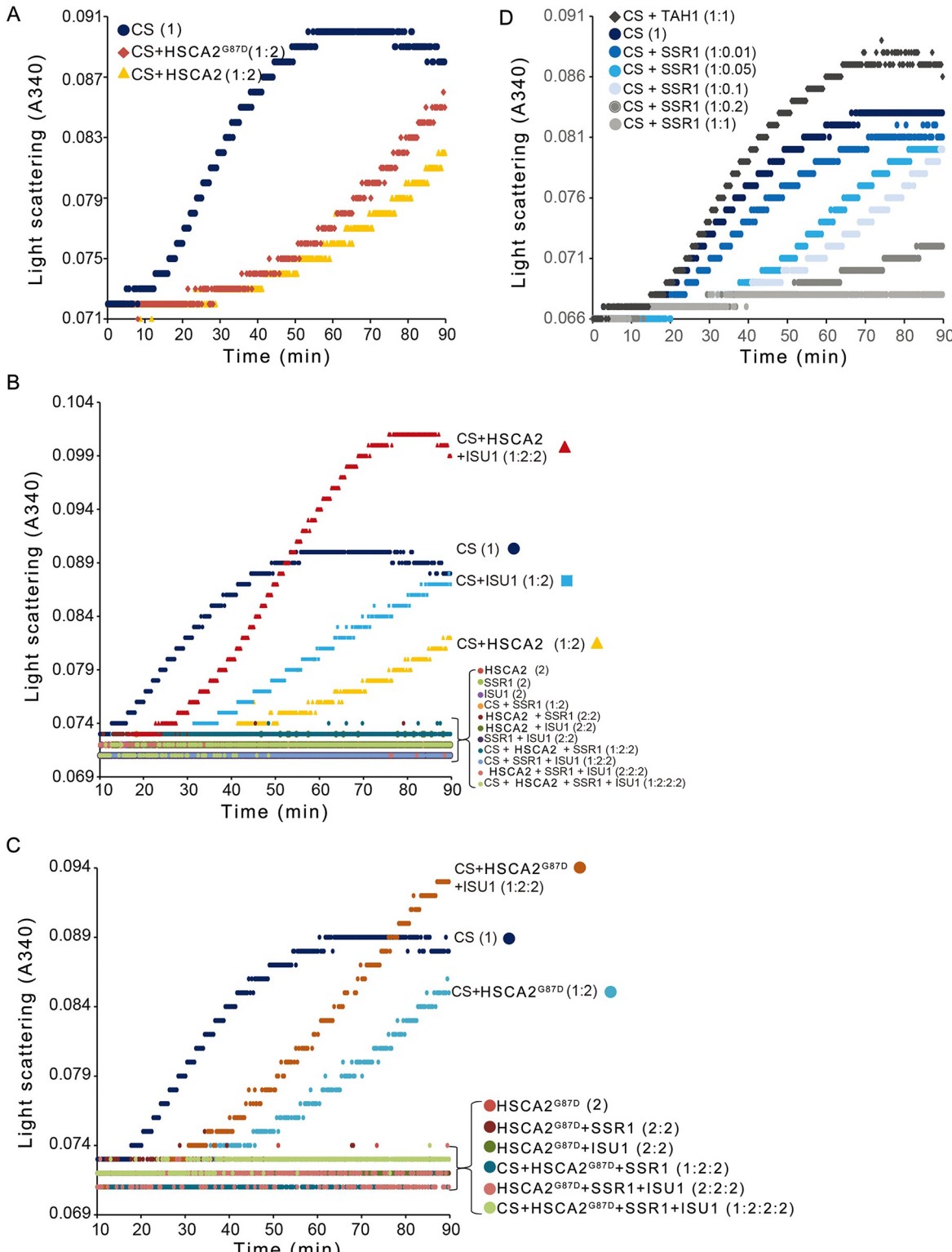

**Fig 4. SSR1 appears strong general molecular chaperone activity.** Heat-induced aggregation of citrate synthase (CS) was performed at 45˚C for 90 min and monitored by increased light scattering at 340 nm. The molecular ratios of CS to tested proteins are indicated within brackets following each protein sample. Control samples without adding CS or those tested protein mixtures that did not show significant protein aggregation are showing curves crowded along the basal line, and therefore shown together without referring individual curve. Only one representative absorbance curve is shown for each tested sample mixture. (*A*) Heat-induced CS aggregation with HSCA2 and

HSCA2[G87D]. (*B*) Heat-induced CS aggregation with HSCA2 in combination with ISU1 and/or SSR1. (*C*) Heat-induced CS aggregation with HSCA2[G87D] in combination with ISU1 and/or SSR1. (*D*) Heat-induced CS aggregation with different amounts of SSR1. TAH1 is a TPR-containing yeast protein and used as a negative control that does not inhibit heat-induced CS aggregation. These tests have been repeated 3 times.

activity of malate dehydrogenase, which is not an iron-sulfur protein, displayed no difference between the wild type and *ssr1-2* (Fig 6D). Additionally, the protein level of ATP5A, a Fe-S cluster-independent subunit of complex V, was higher in *ssr1-2*, while ISU1 exhibited comparable levels between the samples (Fig 6D). Mitochondrial dysfunction often affects mitochondrial membrane potential (MMP). To minimize the interference from chloroplasts, seedlings grown on MS medium in the dark were used for crude mitochondria preparation. A flow cytometry analysis revealed that the MMP was significantly diminished in *ssr1-2* compared to the wild type, reaching a mere 30% of the wild type level. However, in *ssr1-2 sus1* and *ssr1-2 sus2* double mutants, the MMP exhibited a partial restoration, approaching 55% of the wild

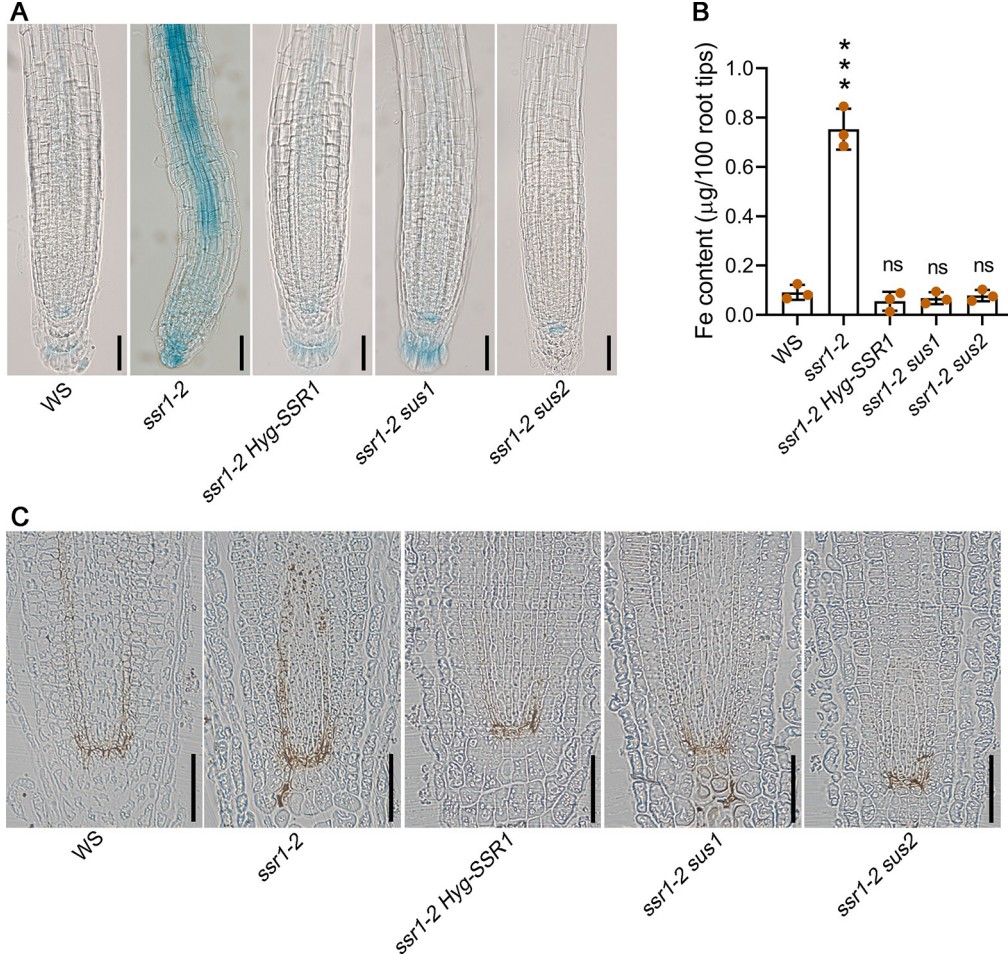

**Fig 5. SSR1 affects iron homeostasis.** (*A*) Perls' prussian blue staining of root tips. (B) Iron content in root tips (about 0.5 cm). (C) Semi-thin (1-mm) longitudinal sections of Perls/DAB-stained root tips. This method allows for the sensitive detection of trace amounts of iron in tissues. The seedlings utilized in Fig 5 were 10-days old and were cultivated on MS medium. Scale bars in 5*A* and 5*C* are 50 μm. In Fig 5B, statistical significance was determined through student *t*-test. The asterisk "***" represents a statistically significant result with *p*-values < 0.001. The symbol "ns" represents a statistical insignificant result with *p*-values < 0.05.

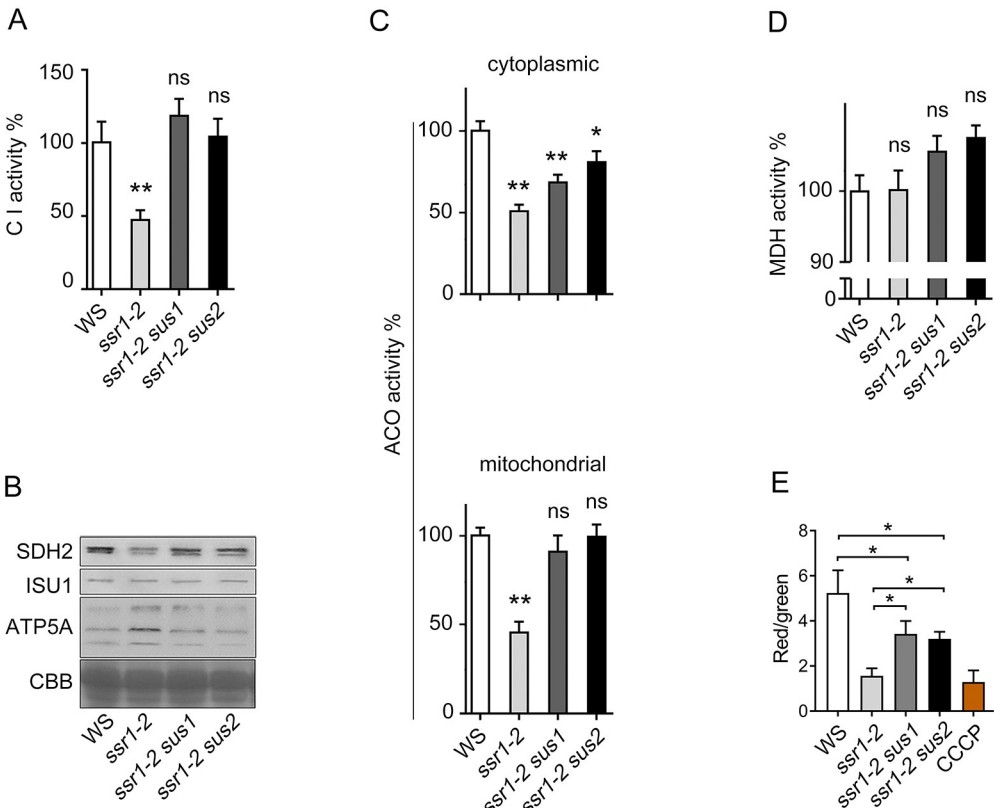

**Fig 6. The activity of representative Fe-S containing enzymes and mitochondrial membrane potential decreased in *ssr1-2* mutant, and were restored by *sus1* or *sus2*.** Mitochondrial complex I (*A*) and aconitase (bottom panel of 6*C*) enzyme activities. Cytosolic aconitase (top panel of 6*C*) and malic dehydrogenase (*D*) enzyme activities. (*B*) Protein expression levels of SDH2, ISU, and ATP5A as detected by corresponding antibodies and the total protein shown by Coomassie brilliant blue (CBB) staining. (*E*) Mitochondrial membrane potential (MMP). CCCP, a mitochondrial oxidative phosphorylation uncoupler, was used as positive control for mitochondria membrane depolarization. Flow cytometry was used for MMP detection, with over ten thousand mitochondria from each sample measured in every replicate. Error bars represent standard deviation (*n* = 3). Statistical significance was determined through student *t*-test. Asterisks indicate a significant difference to the WS or between two designated groups. **$P < 0.01$; *$P < 0.05$. "ns" represents statistically insignificant.

type level (Fig 6E). Taken all data together, it is concluded that *SSR1* is required for Fe-S proteins activity, thus further confirming the role of *SSR1* in the maintenance of mitochondrial electron-transport chain as revealed by analyzing a phenotypically weak mutant *ssr1-1* under the proline treatment [30].

## *SSR1* only exists in plant lineages and may be important for environmental adaptability

It has been shown that *ssr1* mutants are hypersensitive to osmotic stress and proline treatment [29,30], which were believed to cause electron overflow in mitochondrial electron transport chain (mETC) due to decreased mETC activity or elevated electron supply [36,37], and consequently resulted in the burst of reactive oxygen species (ROS) [38]. Actually, ISC machinery is prone to inhibition by ROS under various environmental stresses [39]. Our observations indicated that the *sus1* and *sus2* were unable to effectively rescue the hypersensitive phenotype of the *ssr1-2* mutants to proline treatment, in contrast to their performance under no-stress conditions (Figs 7A, 7B, 1A and 1D). Moreover, *SSR1* was induced by proline, dehydration,

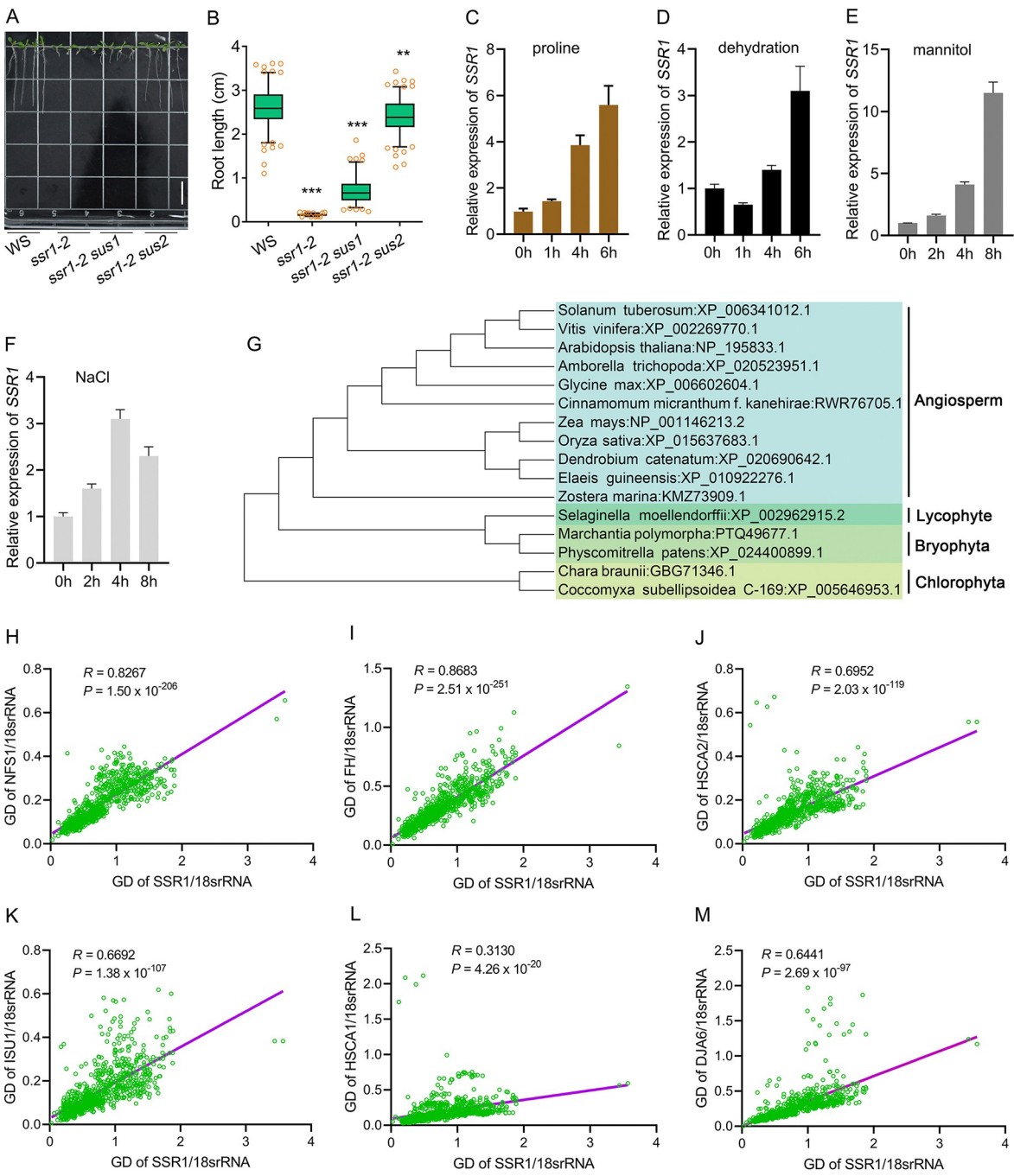

**Fig 7. SSR1 co-evolves with the ISC machinery in plant lineages and may be important for environmental adaptability.** (A, B) Phenotype of 10-day-old seedlings under proline (20 mM) treatment. (C-F) The relative transcript level of *AtSSR1* under different stresses. 10-day-old seedlings were used for treatment with proline (40 mM), mannitol (150 mM), and NaCl (100 mM). For dehydration treatment, detached leaves from 30-day-old plants were placed in a greenhouse for natural dehydration, and the control group was floated in water. The control group was used to normalize the data of the experimental group at each time point. *Actin7* and *eIF4a* were used as the internal reference genes. (G) Phylogenetic tree was constructed using SSR1 protein sequences from representative species at different evolutionary status. (H-M) Pairwise genetic distances (GD) among designated proteins in representative species (S6 Table) are indicated by green circles. *R*, Pearson correlation coefficient; *P*, Significant differences between the Maximum Likelihood trees (one-sided *t*-test).

mannitol, and NaCl treatments (Fig 7C–7F), indicating that *SSR1* may be important for environmental adaptability.

The ISC machinery is evolutionarily conserved from bacteria, fungi to higher plants and mammals. This prompted us to investigate the evolutionary origin of *SSR1* and the evolutionary relationship between *SSR1* and the ISC system. It is notable that *SSR1* is not inherited from microorganisms. Rather, it emerged as a novel gene present only in plant lineages with a distant homolog appearing already in green algae (Fig 7G). Comparison of the evolutionary distances between the sequences of the associated protein families is an efficient way to examine possible protein–protein functional interactions [28]. After alignments of the predicted orthologs and constructions of the phylogenetic trees of NFS1, FH, HSCA1, HSCA2, and ISU1, we found that the phylogenetic trees of SSR1 and proteins of the ISC machinery share similar topologies (S6 Fig and S6 Table), implying that their evolution is highly coordinated. These tree similarities were further quantified by the Tol-Mirror-Tree analysis. Notably, strong correlations (R > 0.600) were observed between the SSR1-NFS1, SSR1-FH, SSR1-HSCA2, and SSR1-ISU1 trees compared to the SSR1-HSCA1 tree (Fig 7H–7M), indicating that SSR1 co-evolves with the ISC machinery and that HSCA2 rather than HSCA1 may be a major component of the ISC machinery.

In order to test whether *SSR1* could play any adaptive roles under stress conditions, *SSR1* was introduced into yeast cells whose ISC machinery doesn't possess SSR orthologs. SSR1-GFP could be detected in the mitochondria of yeast cells (S7A and S7B Fig). However, its expression had an adverse effect on the normal growth of yeast and was unable to enhance the tolerance of yeast to various stress (S7C Fig).

## Discussion

### SSR1 plays a role in ISC machinery by facilitating the interaction between HSCA and ISU

SSR1 is a mitochondrial TPR domain-containing protein and was previously reported to be required for the function of mitochondria electron transport chain [30] and primary root elongation [29]. However, the molecular mechanism underlying the function of *SSR1* remained elusive. In this report, we have pinpointed, by suppressor characterization, that the function of SSR1 in mitochondria as a crucial chaperone-like component to promote the association of HSCA2 with ISU1, with the possibility to facilitate the mitochondrial Fe-S clusters assembly.

It has been well documented that the binding of HSCA to ISU is required for the release of Fe-S clusters from ISU to apo-proteins [16,17,19,20,22]. Our results demonstrated that the optimal binding of HSCA2 to ISU1 is largely SSR1-dependent, as evidenced by the significantly weaker binding observed in the absence of SSR1. However, HSCA2$^{G87D}$ and ISU1$^{T55M}$ could interact well with ISU1 and HSCA2, respectively, thus by-passing the need of SSR1. It appeared that the enhanced affinity between HSCA2 and ISU1 in two suppressor mutants was responsible for suppressing the defect of *ssr1-2*. This was demonstrated by artificially reducing the affinity between HSCA2 and ISU1$^{T55M}$ by certain amino acid substitution in the LPPVK motif of ISU1$^{T55M}$, which render ISU1$^{T55M}$ incapable of suppressing the *ssr1-2* phenotype.

Fe-S clusters are ubiquitous and essential protein cofactors for many proteins engaged in a multitude of processes associated with bioenergetics, metabolism, DNA synthesis and phytohormone biosynthesis [9,33]. Therefore, it was not surprising that *ssr1-2* mutant displayed pleiotropic growth defect, such as short roots, brush-like roots, bushy shoots and growth retardation [29]. Various degrees of growth retardation have been observed in other mutants deficient in the key components of Fe-S cluster assembly system, such as AtNFS1 and AtISU1. The loss of function of *AtNFS1* is lethal [25]. The knockdown of *AtNFS1* expression resulted in a

number of striking visible phenotypes, including dwarf stature, chlorotic spots on the leaves, scalloped edges on new leaves, disorganized inflorescences, and a noticeable increase in axillary shoot development at the axils of both rosette and cauline leaves, resulting in a bushy shoot [25]. The knockdown of *AtISU1* expression resulted in similar phenotypes as that of *AtNFS1* [25]. The dysfunction of the Fe-S clusters biosynthesis machinery also disrupts iron and sulfur homeostasis [33,34]. Genes involved in Fe and S uptakes, assimilation, and regulation were up-regulated in *AtNFS1* overexpressing plants and down-regulated in the knockdown plants [40]. Specifically, the overexpression of *AtNFS1* resulted in elevated Fe and S accumulation, while the knockdown of *AtNFS1* led to reduced Fe and S accumulation [40]. ATM3, an ATP-binding cassette transporter in mitochondria, was proposed to transport some unknown intermediates for Fe-S cluster assembly in cytosol. *ATM3* T-DNA insertional mutant, *atm3-1*, showed dramatically reduced content of abscisic acid and the enzymatic activities of some cytosolic Fe-S proteins, resulting in a pleiotropic growth defect, such as small stature, lower chloroplast numbers and male sterility [41,42]. Strikingly, in contrast to mutants in the yeast and mammalian orthologs, Arabidopsis *atm3* mutants did not display a dramatic iron homeostasis defect and did not accumulate iron in mitochondria [41,42]. Two DnaJ chaperones, DJA6 and DJA5, have recently been identified as essential for chloroplast Fe-S cluster biogenesis. DJA6 and DJA5 proteins are capable of binding iron through their conserved cysteine residues and facilitate iron incorporation into Fe-S clusters via interactions with the SUF apparatus through their J domain [28]. The absence of these two proteins results in severe defects in the accumulation of chloroplast Fe-S proteins, a dysfunction of photosynthesis, a significant intracellular iron accumulation, and growth inhibition [28]. Notably, *ssr1-2* mutants accumulated higher levels of iron in their root tips. The increased iron deposition may be detrimental to the maintenance of the quiescent center and stem cell niche, as the excess ferrous iron may trigger the Fenton reaction, leading to oxidative stress and damage to cellular structures.

It was well-established in microorganisms that HSCB/Jac1 could promote the interaction between HSCA and ISU, and stimulate the release of Fe-S clusters from ISU to apo-proteins [43,44]. The knockout mutants of *AtHSCB* exhibit significantly reduced activities of the Fe-S cluster containing enzymes ACO and SDH [26,35]. However, the activities of the Fe-S cluster-containing enzymes, such as ACO and CI, are not impacted by the dysfunction of GRX15, which is proposed to be the primary carrier of [2Fe-2S] cluster during its release from the ISU scaffold [45]. Conversely, *grx15* primarily influences the activity of lipoyl-dependent proteins [45,46]. Overexpression of *AtHSCB* resulted in the activation of the iron uptake system and iron accumulation in roots, without concomitant transport to shoots. This ultimately led to a reduction in iron content in the aerial parts of plants. By contrast, *hscb* knockdown mutants exhibited the opposite phenotype, with iron accumulation in shoots despite the reduced levels of iron uptake in roots [35]. The *SSR1* knockout mutants have been observed to suppress the expression of iron deficiency response genes and to increase iron accumulation in root tips. The *jac1* yeast mutant is lethal, indicating that *Jac1* is an essential gene in yeast [15]. However, the defective phenotype of *athscb* mutant is relatively mild [26,35]; this suggests the possible existence of functional redundancy genes. Could SSR1 serve as a functionally redundant cochaperone of HSCB in Arabidopsis. HSCB does not exhibit intrinsic chaperone activity and only function as a cochaperone for HSCA [47]. The robust chaperone function of SSR1 and the different role of SSR1 and HSCB in iron homeostasis indicated that SSR1 did not function the same as HSCB. However, in the process of SSR1-assisted binding of HSCA2 to ISU1, it seems more appropriate to call SSR1 as a cochaperone. This aligns with previous reports that the TPR domain binds to the C-terminus of HSP70 [48]. Since SSR1 can bind to and protect

citrate synthase as a chaperone, it may also bind to certain other proteins and perform various biological functions.

## SSR1 may be a plant specific component in ISC machinery for stress adaptation

Unlike most of the other components in ISC system, *SSR1* is not inherited from microorganism. Rather, it emerged as a new gene present only in plant kingdom with a distant homolog appearing already in green algae. Currently, we are not clear about the evolutionary advantage for plant to possess *SSR1*. One clue is that *ssr1* is hypersensitive to osmotic stress [29] and proline treatment [30]. Both of these factors were previously believed to cause electron overflow in the mitochondrial electron transport chain (mETC) due to inhibited mETC activity or elevated electron supply [36,37]. Actually, ISC machinery is prone to inhibition by ROS under various environmental stresses during their sessile life cycle, to which plants must respond and adapt [39]. It has been recently reported that hypoxia can rescue FH/frataxin loss-of-function caused dysfunction of ISC machinery [49]. Although HSCA2[G87D] or ISU1[T55M] can effectively rescue the defect of *ssr1-2*, they did not rescue the sensitive phenotype of *ssr1-2* to proline treatment. We propose that *SSR1* may be required for protecting ISC machinery from stresses, which cannot be completely replaced by HSCA2[G87D] or ISU1[T55M]. Components of Fe-S cluster assembly machineries were recently reported to be robust phylogenetic markers [50]. The complexification of the ISC machinery and expansion of its components are indicative of the capacity of organisms to adapt to the environment [9]. Therefore, the emerging of *SSR1* in plant lineages may confer adaptive advantages on ISC machinery in response to environmental stress. In line with this, *SSR1* was shown to be transcriptionally up-regulated by several abiotic stresses. However, our observations did not indicate that the expression of AtSSR1 in yeast cells could enhance the stress adaptation of yeast. Conversely, the expression of AtSSR1 inhibited yeast growth. We proposed that this may be due to the lack of co-evolution of other components (such as HSCA and ISU) of the ISC machinery in yeast that are compatible with AtSSR1.

In conclusion, we have presented evidence that SSR1 is a new and vital cochaperone-like component for mitochondrial Fe-S clusters biosynthesis in plants.

## Materials and methods

### Plant materials and growth conditions

For seedling growth experiments, surface-sterilized seeds of *Arabidopsis* wild type (WS), transgenic plants and mutants were sown on solid Murashige and Skoog (MS) medium, containing 1% sucrose and 0.25% phytogel, and stratified at 4°C for 2 days in the dark before being transferred to an incubator for germination at 22–24°C, 16-h light/ 8-h dark photoperiod with a light intensity of 110 μmol.m$^{-2}$.sec$^{-1}$. To grow to maturity, 4-day-old seedlings were planted in soil and grown under similar condition as in the incubator. All transgenic plants used in this study were listed in S7 Table.

### Cloning of *SUS1* and *SUS2*

*ssr1-2 sus1* and *ssr1-2 sus2* were backcrossed twice with the *ssr1-2* parent to obtain F1 plants. The F1 plant was self-pollinated to obtain F2 progeny, which were further planted and harvested individually. The root length of the F3 population was investigated. Populations exhibiting no apparent segregation of root lengths were deemed homozygous for the *SUS1* and *SUS2* loci. The 30 F3 populations with the longest root length were selected, and an equal number of plants from each population was mixed to extract DNA to construct a long-root pool. A total

of 30 F3 populations exhibiting the same root length as *ssr1-2* were selected, and an equal number of plants from each population was mixed to extract DNA to construct a short-root pool. DNAs isolated from *ssr1-2* and two pools were subjected to genome re-sequencing. Super Bulked Segregant Analysis [51] was used to screen for candidate mutation sites that are correlated with the root length phenotype (S2 and S3 Tables). Trait-related candidate regions were selected based on the differences in genotype frequency in mixed pools, and were calculated using the Euler distance (ED) correlation algorithm. The quantile-based method was used to select the threshold value. All fitted values were sorted from the smallest to the largest, and the SNP labeled ED value greater than 99% or 99.9% was selected as the threshold value during screening. Subsequently, the genes with SNPs that were most correlated with root length were subjected to further analysis based on their functional annotation and prediction of protein subcellular localization. The five most likely candidate genes were cloned from each suppressor mutant and introduced into *ssr1-2* for genetic complementation (S2 and S3 Tables).

## Plasmids construction and plant transformation

All constructs and primers used in this study were listed in S8 Table. Briefly, for genetic complementation, the genomic sequences for *sus1*, *sus2*, *sus5* and *sus6*, which encode HSCA2$^{G87D}$, ISU1$^{T55M}$, ISU1$^{G106D}$ and ISU1$^{A143T}$ respectively, were amplified and cloned from corresponding suppressor mutants. To generate HSCA1$^{G82D}$ from wild type HSCA1, the *HSCA1* coding sequence (CDS) was amplified in two halves separately, and then fused together by overlap extension PCR with the simultaneous introduction of the mutation. All complementation constructs were based on the binary vector pCAMBIA1300 and pCAMBIA1300-super-Flag. For protein expression in *E. coli*, the coding sequences of *HSCA2*, *SSR1* and *ISU1*, with or without mitochondrial targeting peptide coding sequences, were cloned into vector pRSET and fused with the Myc-tag. Amino acid substitution mutants around the LPPVK motif of ISU1 were also generated with overlap extension PCR. To generate constructs, transfected and transgenic plants for co-IP analysis, the promoter and CDS of *HSCA2* was amplified and first inserted into pBI121-cMyc. Then the Myc-tagged expression cassette was amplified and inserted into pCAMBIA1300. The pSSR1-Flag was described previously [29]. For BiFC assays, the CDS of *SSR1*, *HSCA2* and *ISU1* from wild type were inserted into pE3242 or pE3228 to generate either nVenus-tagged or cCFP-tagged constructs, respectively. See Lee's work for more details about vectors pE3228 and pE3242 [52].

Whenever needed, *Arabidopsis* plants were transformed with Agrobacteria-mediated floral dipping method [53] or the protoplasts were transfected with expression constructs as described [54]. The trangenic plants were screened on kanamycin- or hygromycin B-containing MS medium depending on the vector used. The integration of the transgene was confirmed by PCR.

## Protein expression and purification from *E. coli*

His-tagged protein expression and purification from *E. coil* was carried out as described previously [22]. Briefly, BL21(DE3) bacterial strains with respective constructs were cultured in LB liquid medium at 37˚C to OD$_{600nm}$≈0.5, and then induced with IPTG at a final concentration of 1 mM for 6 h at 28˚C. Cells were harvested, re-suspended in buffer A [20 mM Tris–HCl, 200 mM NaCl, 30 mM imidazole and 1 mM phenylmethylsulfonyl fluoride (PMSF), pH 7.4] and then disrupted by sonication. The suspensions were centrifuged at 10,000 ×g for 15 min at 4˚C. The supernatants of the His-tagged proteins were purified using His-tag protein purification columns (Cytiva: HisTrap HP). The recombinant proteins were eluted with buffer B (500

mM imidazole in buffer A). In order for proteins to be utilized in in vitro chaperone activity assays, the eluants were subjected to further application of size exclusion chromatography with the Superdex 75 or Superdex 200 column, utilizing the ÄKTA Purifier 10 FPLC system (GE Healthcare). The *E. coli* lysates containing the recombinant proteins without His-tag were directly used for the pull-down assays.

## Protein-protein interaction assays

Pull-down assay was carried out as described previously [22]. For purified proteins, His-tagged proteins and tag-excised proteins were incubated together equally (about 10 μg of each protein in a total volume of 200 μL). The incubation buffer is formulated as: 50 mM tris-HCl (PH7.4), 50 mM NaCl, 10 mM KCl, 0.1% Tween-20, and cocktail protease inhibitors. For the recombinant proteins without His-tag, about 10 μg purified His-tagged HSCA2, HSCA2$^{G87D}$ or SSR1 was incubated with BL21(DE3) cell lysates expressing SSR1, ISU1 or ISU1$^{T55M}$ without His-tag for 1 hour at 4°C. Next, 20 μL Ni Sepharose pre-equilibrated with buffer A [20 mM Tris–HCl, 200 mM NaCl, 30 mM imidazole and 1 mM phenylmethylsulfonyl fluoride (PMSF), pH 7.4] were added and incubated for another 2 hours at 4°C. After washing twice with buffer A, the proteins binding to Ni-NTA Sepharose (GE Healthcare: Ni Sepharose 6 Fast Flow) were eluted by appropriate volume of buffer B (500 mM imidazole in buffer A), and analyzed by Western blot.

The first dimension of BN-PAGE and second dimension of SDS-PAGE were performed as described by Wittig et al. [55]. Briefly, Different designated combinations of proteins were mixed (about 10 μg of each protein in a total volume of 200 μL) and incubated at 4°C for one hour, after which they were subjected to a BN-PAGE assay. The incubation buffer is formulated as: 50 mM tris-HCl (PH7.4), 50 mM NaCl, 10 mM KCl, 0.1% Tween-20, 100 μM ATP, and cocktail protease inhibitors. Cathode buffer and anode buffer were the same as described by Wittig et al. [55]. A gradient separation gel (5%-13% acrylamide) was employed for the first dimension of BN-PAGE, while a 10% acrylamide separation gel with β-mercaptoethanol was utilized for the second dimension of SDS-PAGE.

BiFC assays were performed via transient expression in Arabidopsis mesophyll protoplasts as previously described [54]. Co-IP assays from transgenic plants were performed with C-Myc Isolation Kit (130-091-123) and DYKDDDDK (Flag) Isolation Kit (130-101-591) from Miltenyi Biotec. Briefly, 12-day-old transgenic seedlings containing *HSCA2$_{pro}$:HSCA2-Myc* and *35S$_{pro}$:SSR1-Flag* were ground to fine powder in liquid nitrogen. About 500 mg powder was transferred to 2 mL Eppendorf tube, before 1.5 mL pre-cooled lysis buffer (50 mM Tris-HCl, pH7.9, 120 mM NaCl, 10% glycerol, 10 mM DTT, 5 mM EDTA, 1% PVP, 1% NP-40, 1 mM PMSF, protease inhibitor cocktail) was added and mixed well with the powder. The homogenates were incubated for 30 minutes on ice with occasional mixing and then centrifuged for 10 minutes at 10,000 ×g at 4°C to get rid of the debris. Total input samples were taken from the supernatant, and the remaining supernatant was transferred to a fresh 1.5 mL tube with 50 μL antibody and was incubated on ice for 40 minutes. Place μ column (130-042-701) in the magnetic field of the μMACS Separator and prepare the μ column by applying 200 μL lysis buffer on the column. Transfer the supernatant into the μ column and let the lysate run through. Rinse the column with 4×200 μL of lysis buffer and 1×100 μL wash buffer (20 mM Tris-HCl, pH 7.5). Pre-heated elution buffer (from Kit) was subsequently added to elute the immunoprecipitated, which was used for Western blot.

The antibodies used for Western blot were purchased from Abcam (anti-ISU: ab154060; anti-SDH2: ab154974; anti-ATP5A: ab14748) and Sigma-Aldrich (anti-Myc: M4439; anti-Flag: F3165).

### In vitro chaperone activity assay

All tested proteins and citrate synthase (CS, Sigma, C3260) was dialyzed in 20 mM HEPES-KOH, pH 7.5, 150 mM KCl, 10 mM $MgCl_2$ before being used for the heat-induced aggregation assay. CS (500 nM) was prepared in a final volume of 150 μL 20 mM HEPES-KOH (pH 7.5), 2.8 mM β-mercaptoethanol with different amounts of tested proteins. The mixtures were added to a 96-well microplate and heated at 45°C. Light scattering at 340 nm was monitored at 45°C in a Synergy 4 spectrophotometer (BioTek) for 90min. Control measurements were performed with purified test proteins alone in the absence of CS and a $His_6$-tagged yeast TPR-containing Tah1 protein expressed and purified from *E. coli* was also used as a negative control [56].

### Isolation of mitochondria

Mitochondria was isolated as previously described [57] with minor modification using 10-day-old seedlings grown on solid MS medium. Briefly, plant tissue was homogenized in Mito Homogenization buffer (0.35 M mannitol; 30 mM MOPS (pH7.3); 1 mM EDTA (pH8.0); adding 0.2% (w/v) BSA, 0.6% (w/v) polyvinylpyrrolidone-40 and 0.3% (w/v) ascorbic acid in cooled autoclaved buffer), filtrated through one layer of Miracloth and two layers of muslin to remove starch, cell debris, and unbroken cells, and centrifuged at 5000 x g for 5 min. The supernatant was gently transferred into fresh centrifuge tubes and centrifuged at 25000x g for 15 min. The pellet was gently resuspended in 2 mL Mito Homogenization buffer, loaded on a cooled Percoll gradient in 10 mL Oak ridge tubes (bottom to top, 2 mL of 45% Percoll, 3 mL of 27% Percoll, 2 mL of 21% Percoll in 0.25 M sucrose, 10 mM MOPS (pH7.3), 0.2% (w/v) BSA), and centrifuged at 25000 x g for 30 min. Crude mitochondria will form a white/pale band between 45% and 27% Percoll or in 45% Percoll toward the bottom.

### Enzyme activity and MMP analysis

Crude mitochondria from seedlings grown in light/dark photo-period were used for analysis of enzymatic activity for complex I (Catalog number: FHTA-2-Y), aconitase (Catalog number: ACO-2-Z), and malate dehydrogenase (Catalog number: NMDH-2-Y) with kits from Suzhou Comin Biotechnology Co. Ltd according to the instruction manuals.

Crude mitochondria from dark-grown seedlings were used for mitochondrial membrane potential (MMP) measurement using kit from Beyotime (Catalog number: C2006) with JC-1, a fluorescent indicator. Flow cytometry (Beckman: Moflo XDP) was used to detect the MMP. The ratio of red fluorescence to green fluorescence can be utilized to assess the potential of the mitochondrial membrane. A smaller ratio is indicative of a mitochondrial membrane that is in a depolarized state.

### Histochemical Fe staining

The Fe-specific Perls staining was adapted from Muller et al [58]. The seedlings were washed twice with distilled water ($ddH_2O$) and incubated in a 10% neutral formaldehyde solution for two hours. For the intensification of the 3,3'-diaminobenzidine (DAB) reaction and the preparation of semi-thin (1 mm) sections, the roots were fixed with a 4% glutaraldehyde solution for four hours. Subsequently, the seedlings were washed with $ddH_2O$ twice and transferred to a staining buffer [4%(v/v) HCl, 4% (w/v) K-ferrocyanide] for a 30 min incubation at room temperature. The seedlings were then washed ($dH_2O$) and incubated (1 hour) in methanol containing 10 mM sodium azide and 0.3% (v/v) $H_2O_2$. Following a wash with 100 mM Na-phosphate buffer (pH 7.4), the plants were incubated for up to 30 min in the same buffer

containing 0.025% (w/v) DAB and 0.005% (v/v) $H_2O_2$. The reaction was stopped by washing with water and incubation with chloral hydrate (1g/mL in 15% glycerol). Perls/DAB-stained roots and sections were analyzed on a Zeiss Axio Imager M2.

## Measurement of Fe content

The Fe content was quantified via ICP-MS (PerkinElmer: NexION 350). The root tips (0.5 cm) of 10-day-old seedlings were harvested and washed twice in $ddH_2O$. The root tips were finely ground in a 1.5 mL tube. Then, 1 mL of 2% nitric acid was added to it and incubated for one day. Subsequently, the samples were subjected to centrifugation at 12,000 g, after which the supernatant was collected for subsequent analysis. Each sample comprised 100 root tips, and three replicates were prepared.

## Gene expression analysis

Primers used for real-time qPCR and T-DNA insertion mutant detection were listed in S9 Table.

## Evolutionary analysis

The amino acid sequence of *Arabidopsis* SSR1 was used to search for orthologous proteins from microorganism to higher organisms using UniPro BLAST and NCBI BLASTP. The similarity of ISC components between *E. coli* and *Arabidopsis* (ISU1/2/3, HSCA1/2, HSCB, ADX1/2 and FH) was used as a reference for threshold value (*E*-value less than $1*10^{-4}$; identity larger than 20%) to filter out the returned proteins displaying low similarity to SSR1. Then, the candidates from different representative species were used as query to search in *Arabidopsis* database with the same threshold value. If *Arabidopsis* SSR1 was returned as subject, the candidate would be selected as orthologs.

Maximum Likelihood phylogenetic trees of SSR1 orthologs, NFS1 orthologs, FH orthologs, HSCA2 orthologs, ISU1 orthologs and HSCA1 orthologs were constructed using MEGA 7.0.26 [59] with parameters of the Jones-Taylor-Thornton (JTT) model, complete deletion and 1,000 replicates of bootstrap.

## Accession numbers

Sequence data for genes and proteins presented in this article can be found in the Arabidopsis Genome Initiative of GenBank/EMBL database under the following accession numbers: *SSR1* (AT5G02130), *HSCA2* or *mtHSC70-2* (At5g09590), *ISU1* (At4g22220), *HSCA1* or *mtHSC70-1* (AT4G37910), *HSP70b* (AT1G16030), *cpHSC70-2* (AT5G49910), *HSP70-1* (AT5G02500), *HSP70T-1* (AT1G56410), *cpHSC70-1* (AT4G24280), *HSP70T-2* (AT2G32120), *NFS1* (AT5G65720), *FH/frataxin* (AT4G03240), *Actin 7* (AT5G09810), *eIF4a* (AT3G19760).

## Supporting information

**S1 Table. Root length of F2 seedlings.**
(XLSX)

**S2 Table. BSA-seq result of *sus1*.**
(XLSX)

**S3 Table. BSA-seq result of *sus2*.**
(XLSX)

**S4 Table. Summary of identified suppressors.**
(XLSX)

**S5 Table. Contents of different metal ions.**
(XLSX)

**S6 Table. Species and the accession numbers for sequences used for the evolutionary analysis of SSR1, NFS1, FH, HSCA1, HSCA2 and ISU1 orthologs.**
(XLSX)

**S7 Table. Transgenic plants used in this study.**
(XLSX)

**S8 Table. Plasmid constructs and primers used in this study.**
(XLSX)

**S9 Table. Primers for real-time qPCR or plant lines genotyping.**
(XLSX)

**S1 Fig. *sus1* and *sus2* may be a dominant and a semi-dominant mutation, respectively.** (*A*) The root length of the F1 population of *ssr1-2 sus1* x *ssr1-2* and *ssr1-2 sus2* x *ssr1-2* was compared with that of their respective parents, *ssr1-2 sus1* and *ssr1-2 sus2*. (*B*) The two F1 populations were self-pollinated to generate F2 seeds. The primary root length of the F2 seedlings (10-days-old) were measured. The original root length data are shown in S1 Table. Three biological repeats were performed and shown in differently colored lines. The segregation ratio of the root length phenotype indicates that *sus1* is a dominant mutation and *sus2* is a semi-dominant mutation.
(TIF)

**S2 Fig. Several point mutations of HSCA and ISU1 can rescue the defect of *ssr1-2*.** (*A*) *sus5* and *sus6* mutant genes were cloned, designated as *ISU1^{G106D}* and *ISU1^{A143T}*, respectively, and re-transformed back to *ssr1-2* plants. In transgenic lines, the transgenes are associated with a hygromycin resistant gene and therefore designated as *Hyg-ISU1^{G106D}* or *Hyg-ISU1^{A143T}*. It should be noted that *ssr1-2 Hyg-ISU1^{G106D}* and *ssr1-2 Hyg-ISU1^{A143T}* still contain the wild type *ISU1* allele. (*B*) HSCA1^{G82D} is a homolog of HSCA2^{G87D}, and it can rescue the phenotypes of *ssr1-2*. The naming rule of transgenic line was similar to 1*A*. The red asterisk labeled amino acid residue G is conserved among various member of HSP70 family except HSP70T-2. (*C*) HSCA2 loss of function mutation has no effect on root length. Primers hsca2-F and hsca2-R were used for the amplification of *HSCA2* genome fragment, and hsca2-F and LB were used to detect the T-DNA insertion. (*D*) The single mutant *sus1* and *sus2* have similar root length with the wild type. *sus1* and *sus2* single mutants were isolated after back-crossing *ssr1-2 sus1* and *ssr1-2 sus2* with the wild-type. For (*A-D*), representative seedlings grown for 10 days and average root length of analyzed lines were shown. Error bars represent standard deviation from 20 (*A*), 30 (*B*), 60 (*C*) and 100 (*D*) seedlings on primary roots. Scale bars = 1 cm. The statistical significance of the results was determined through the use of a student *t*-test. Variations were considered significant if P <0.05(*), 0.01(**) or 0.001(***).
(TIF)

**S3 Fig. SSR1 interacts with HSCA2 and ISU1 and promotes HSCA2-ISU1 association.** In vitro pull-down assays were conducted using Ni-NTA Sepharose with His-tagged proteins expressed and purified from *E. coli* as baits. The recombinant proteins SSR1-Myc, ISU1-Myc and ISU1^{T55M}-Myc were expressed in *E. coli* and the protein lysate was directly used for co-incubation with His-tagged proteins. Co-purified proteins were detected with anti-Myc

antibody. The immunoblotting signals of ISU1-Myc or ISU1$^{T55M}$-Myc were quantified by ImageJ and relative intensities were shown under respective lanes. The mitochondrial targeting peptide coding sequences of HSCA2 and ISU1 were removed for protein expression in *E. coli*.
(TIF)

**S4 Fig. The co-existence of His-SSR1-Myc, His-HSCA2-Myc, and His-ISU1-Myc proteins in a complex has been observed.** Different designated combinations of proteins were incubated at 4°C for one hour, after which they were subjected to a BN-PAGE assay. Subsequently, the BN-PAGE lanes were incised and rotated to a 90-degree angle for SDS-PAGE assay. Finally, the signal of each protein was detected by anti-Myc antibody. The three proteins could be distinguished based on their respective band sizes. The mitochondrial targeting peptide coding sequences of HSCA2, SSR1 and ISU1 were removed for protein expression in *E. coli*.
(TIF)

**S5 Fig. General chaperone activity assays of purified HSCA2 and HSCA2$^{G87D}$.** Heat-induced aggregation of citrate synthase (CS) was performed at 45°C for 90min with different amount of purified test proteins. The molecular ratios of CS to tested proteins are indicated following each protein sample.
(TIF)

**S6 Fig. Phylogenetic trees of SSR1 protein and the ISC machinery components from green algas to higher plants.** Maximum Likelihood phylogenetic trees of SSR1 orthologs, NFS1 orthologs, FH orthologs, HSCA2 orthologs, ISU1 orthologs and HSCA1 orthologs were constructed using MEGA 7.0.26 with parameters of the Jones-Taylor-Thornton (JTT) model, complete deletion and 1,000 replicates of bootstrap. The species and the accession numbers of sequences are shown in figure and listed in S6 Table.
(TIF)

**S7 Fig. The ectopic expression of *AtSSR1* in yeast results in a growth inhibition.** (*A*) Laser confocal microscopy was employed to observe the expression of GFP and SSR1-GFP in yeast cells. (*B*) The yeast cells expressing GFP or SSR1-GFP were exposed to different stress conditions. (*C*) Subcellular localization of SSR1-GFP and mSSR1-GFP in yeast cells. mSSR1 represents the deletion of the N-terminal mitochondrial localization signal peptide. Scale bars represent 5 μm.
(TIF)

**S1 Dataset. The predicted interaction models between HSCA2/HSCA2$^{G87D}$ and ISU1/ISU1$^{T55M}$ using AlphaFold3.** In the cartoons, we label HSCA2 in blue, ISU1 in green, LPPVK amino acid sequence in gray, and Gly87 of HSCA2 and Thr55 of ISU1 in red.
(RAR)

**S1 Appendix. Resource for Fig 1.**
(XLSX)

**S2 Appendix. Resource for Fig 2A.**
(PPTX)

**S3 Appendix. Resource for Fig 2C.**
(PPTX)

**S4 Appendix. Resource for Fig 3B.**
(XLSX)

**S5 Appendix. Resource for Fig 4.**
(XLSX)

**S6 Appendix. Resource for Fig 6.**
(XLSX)

**S7 Appendix. Resource for Fig 7B.**
(XLSX)

## Acknowledgments

We thank Prof. Hongzhi Kong from Institute of Botany, Chinese Academy of Sciences, for his valuable suggestions on phylogenetic tree construction. We also extend our thanks to Dr. Meijing Li from the Shenzhen Medical Academy of Research and Translation for her assistance with the simulation and analysis of protein-protein interactions.

## Author Contributions

**Conceptualization:** Xuanjun Feng, Rongmin Zhao, Xuejun Hua.

**Data curation:** Xuanjun Feng, Rongmin Zhao, Xuejun Hua.

**Formal analysis:** Xuanjun Feng, Yue Hu, Tao Xie, Huiling Han, Diana Bonea, Lijuan Zeng, Jie Liu, Wenhan Ying, Bona Mu, Yuanyuan Cai, Min Zhang.

**Funding acquisition:** Xuanjun Feng, Yanli Lu, Rongmin Zhao, Xuejun Hua.

**Investigation:** Xuanjun Feng, Yue Hu, Tao Xie, Huiling Han, Diana Bonea, Lijuan Zeng, Jie Liu, Wenhan Ying, Bona Mu, Yuanyuan Cai, Min Zhang.

**Methodology:** Xuanjun Feng, Yue Hu, Tao Xie, Huiling Han, Diana Bonea, Lijuan Zeng.

**Project administration:** Xuanjun Feng, Rongmin Zhao, Xuejun Hua.

**Resources:** Xuanjun Feng, Yanli Lu, Rongmin Zhao, Xuejun Hua.

**Supervision:** Xuanjun Feng, Xuejun Hua.

**Validation:** Xuanjun Feng, Yue Hu, Tao Xie, Huiling Han, Diana Bonea, Wenhan Ying, Bona Mu, Yuanyuan Cai.

**Visualization:** Xuanjun Feng, Yue Hu, Tao Xie, Huiling Han, Diana Bonea, Bona Mu, Rongmin Zhao, Xuejun Hua.

**Writing – original draft:** Xuanjun Feng, Rongmin Zhao, Xuejun Hua.

**Writing – review & editing:** Xuanjun Feng, Yue Hu, Tao Xie, Huiling Han, Diana Bonea, Lijuan Zeng, Jie Liu, Wenhan Ying, Bona Mu, Yuanyuan Cai, Min Zhang, Yanli Lu, Rongmin Zhao, Xuejun Hua.

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
