## [Decision Letter · Decision Letter 0]

8 Sep 2024

Dear Dr Feng,

Thank you very much for submitting your Research Article entitled 'The plant-specific chaperone SSR1 affects root elongation by modulating the mitochondrial iron-sulfur cluster assembly machinery' to PLOS Genetics.

The manuscript was fully evaluated at the editorial level and by independent peer reviewers. The reviewers appreciated the attention to an important problem, but raised some substantial concerns about the current manuscript. Based on the reviews, we will not be able to accept this version of the manuscript, but we would be willing to review a much-revised version. We cannot, of course, promise publication at that time.

If you decide to revise the manuscript for further consideration at PLOS Genetics, please aim to resubmit within the next 60 days, unless it will take extra time to address the concerns of the reviewers, in which case we would appreciate an expected resubmission date by email to plosgenetics@plos.org.

To resubmit, log into your Editorial Manager account and select the option 'Revise Submission' in the 'Submissions Needing Revision' folder.

We are sorry that we cannot be more positive about your manuscript at this stage. Please do not hesitate to contact us if you have any concerns or questions.

Yours sincerely,

Nicolas Rouhier

Guest Editor

PLOS Genetics

Claudia Köhler

Section Editor

PLOS Genetics

Reviewer's Responses to Questions

**Comments to the Authors:**

Reviewer #1: This interesting work proposes that the plant-specific protein SSR1 acts as a molecular chaperone regulating the mitochondrial ISC machinery in response to environmental stress. The authors initially identify two suppressors of a ssr1-2 plant which is phenotypically characterized by short root length. Necessary controls have been made to verify the suppressor effects (but since I am not a plant geneticist, I cannot fully evaluate this part). The suppressors are encoded by the HSCA2 and ISU1 genes, both known to be involved in FeS protein formation. Interactions between SSR1 and the two ISC proteins are shown by BiFC (please improve quality of data presentation; I mostly see a black picture) and Co-IP studies. The latter also suggest that SSR1 supports a slightly better interaction between ISU1 and HSCA2. By introducing mutations into the LPPVK motif of ISU1 (known to be needed for HSP70 interaction from studies in bacteria and yeast), it is shown that SSR1 can stabilize their interaction to some extent, but only when these proteins are completely unable to bind to each other. The authors then tested a potential chaperone activity of SSR1 by preventing the aggregation of a model protein (CS). These studies suggest that SSR1 indeed can prevent aggregation of CS. Why is this effect not smaller when ISU1 and HSCA2 are both present since under these conditions these proteins should sequester SSR1 and thereby prevent its chaperone activity? Next, the authors find that ssr1-2 roots accumulate iron because of disturbed iron regulation (using various methods). The authors do not explain why specifically root tips take up more iron. Please, address this issue. Further, why is the iron accumulation higher around the quiescent center and the stem cell niche of the root? All the effects are again suppressed by sus1/2. Since all these data suggest that Ssr1 may play a novel role in plant mitochondrial FeS protein synthesis, the authors measure FeS protein activities (mito and cyto Aco, CI) and find them diminished in ssr1-2 and complemented by Sus1/2. Gene expression of Ssr1 is increased by proline and other measures, yet the precise reason for these effects remains obscure, and may need extra work. Finally, the authors suggest that SSR1 co-evolved with the ISC system, yet is present only in the plant lineage. Overall, the authors provide evidence for a new plant-specific member of the mitochondrial ISC machinery. The data suggest that the Ssr1 protein may act similarly to the J protein HSCB interacting with HSCAs. This aspect is not properly addressed in the manuscript and needs further attention. How does Ssr1 differ from the plant mito HSCB? The discussion of these aspects leaves the reader somewhat puzzled. A few other issues have to be resolved before publication can be recommended.

Comments:

1. Introduction: The Eukarya-specific CIA system should be mentioned as a biogenesis system, since it is known for many years.

2. Introduction, line 57: Also human cells have been widely used to study FeS protein assembly. Please add.

3. Lines 61-62: This statement is certainly wrong: “Fdx/Yfh1 was proposed to supply iron to scaffold IscU/Isu1 62 [12].” The paper does not mention CyaY (= bacterial Yfh1) and shows that Fdx plays a role in cluster fusion. Please read the literature and cite carefully.

4. Line 66 f: “...facilitate the release of Fe-S cluster from IscU/Isu1 to apo-proteins.” Not really correct. Both in bacteria and mitochondria a glutaredoxin is used as a trafficking factor. Please amend these sentences, and also add the DnaJ function, because that is at least as important as that of the Hsp70.

5. The reader is left alone with the function and localization of NFS2 and ABA3 (both are known). Please mention.

6. What is CK in Fig. S7B?

7. The entire manuscript should be checked for typos (there are quite a few).

Reviewer #2: Autonomous assembly of iron-sulfur clusters in mitochondria is essential for supplying all mitochondrial iron-sulfur proteins with their respective clusters. The machinery for this cluster assembly in plants is generally closely related to the machinery in non-plant organisms like e.g. yeast and human cells with orthologous proteins for all key steps of cluster assembly. Therefore, it is very interesting to see here that a new plant-specific component of the cluster assembly machinery is proposed. In a suppressor screen conducted on the mutant ssr1-2, which is characterized by a short and swollen root phenotype, two of the isolated suppressors are mutant variants of the well-established cluster assembly core components ISU1 and HSCA2. While six (?, the number seems to vary in different parts of the text causes some confusion) other potential suppressors could not be validated (or were not characterized further), the mutants sus1 (HSCA2G87D) and sus2 (ISU1T55M) were analyzed in some more depth. These proteins are known elements of the mitochondrial iron-sulfur cluster assembly machinery and known to be involved in the release of newly assembled clusters to apoproteins. It is intriguing that the subcellular localization of these candidates matches the localization of SSR1 that was reported earlier. SSR1 contains a tetratricopeptide domain and is shown here to act as a chaperone that is capable of preventing citrate synthase from heat-induced denaturation. Based on the appearance of ISU1 and HSCA2 mutants in the suppressor screen and further presented results, SSR1 is also considered as a protein that helps establishing close contact between ISU1 and HSCA2 by which it thus may act as a chaperone or co-chaperone in FeS cluster release.

Given that FeS clusters are essential for multiple processes in mitochondria and particularly for energy metabolism, the short root phenotype in ssr1 mutants could be just pleiotropic. The observed decrease in auxin transporters, the transcription factor WOX5 and the amount of auxin may thus be simply a mere secondary consequence of other defects rather than based on a causal link fo FeS cluster assembly in mitochondria. This possibility should be taken into consideration before speculating too much about the effects of FeS cluster deficiency on auxin metabolism and signalling.

To prove the suspected role of SSR1 in FeS cluster supply to the respective apoproteins, activities of complex I in the respiratory chain, cytosolic and mitochondrial aconitases as well as malate dehydrogenase were measured. The provided description of the respective methods is very superficial and by far insufficient for others to repeat the experiments. E.g. the identity of the used kits is not provided and now information on the exact parameter that was measured is given. For proper evaluation of the presented data it is also necessary to show the raw data in supplemental figures rather than presenting only a bar chart for enzyme activities. The reason for this is that at least the measurement of aconitase activities is often prone to artefacts and even more so when the activity for isoforms in different subcellular compartments are to be measured. The authors measured the expected differences and effects similar to what has been described in the past. This may not be completely wrong, but it does not fully match with more recent data from Moseler et al., 2021 and Ströher & Millar, 2016 (both Plant Physiology) who both independently showed that the prime consequence of defects in FeS cluster transfer is actually in protein lipoylation with the respective metabolic consequences. The observed differences in the mitochondrial membrane potential might be a highly pleiotropic effect and thus not sufficient to support the presumed function of SSR1. It is also not clear why technically ambitious isolation of intact mitochondria was done rather than using conventional stains that label mitochondria in live cells dependent on their membrane potential. Furthermore, it is shown that the abundance of SDH2 in ssr1-2 decreases (and partially recovers with expression of sus1 and sus2). Given that aconitase is well known to become instable without FeS clusters it would have been great to show data on the abundance of aconitases as a complement for the activity measurements.

Protein-protein interactions for ISU1, HSCA2 and SSR1 led to the hypothesis that SSR1 is involved in FeS cluster transfer from ISU1 to recipient proteins. The principle idea of an involvement of another chaperone or co-chaperone is not completely new as other chaperones are involved in this process in yeast and human cells. This is also indicated in the text. The finding that point mutations in either ISU1 or HSCA2 can at least partially compensate for the loss of SSR1 is novel and highly interesting. However, for better mechanistic understanding it would be helpful to further explore the effects of the respective mutations on the protein-proteins interactions in more details. Why exactly would the mutation T55M in ISU1 improve its interaction with HSCA2? The same question applies to G87D in HSCA2. Would structural modelling allow to deduce the respective impacts on the interactions?

If SSR1 indeed acts as a chaperone and apparently does interact with citrate synthase as the first tested candidate protein already, it cannot be excluded that it also binds to multiple other proteins as well. This ought to be critically assessed in the discussion. Similarly, it also should be discussed that TPR domain-containing proteins have been reported earlier to interact with Hsp70 and help them as co-chaperones in their function.

Additional points:

l. 60: To avoid confusion with other forms of FeS clusters, it should be considered to explicitly mention [2Fe-2S] clusters here (and elsewhere in the text).

l. 89: ‘TPR’ needs an explanation on first mention

l. 94-94: These two sentences appear partially redundant.

l. 101-102: The very same information was provided a few lines earlier in the Introduction already.

l. 105: here ‘a dozen identified suppressors’ are indicated, while the Abstract refers to only eight suppressors are mentioned (l. 26). Please clarify!

l. 109-116; Fig. 1B: The effect of the ssr1-2 mutation on auxin transport and signalling is very likely highly pleiotropic (as discussed in l. 439-443) and may be merely related to recovery of root growth in the suppressor mutants rather than functionally linked to the mutation. These observations are at best complementary and could be transferred to the supplement without losing any key message. The discussion of a possible decrease in aldehyde oxidase activity due to deficiency in supply of FeS clusters would need a solid experimental base to avoid too much speculation. In this context, see also general comments above on prime metabolic effects of deficiencies in FeS cluster supply.

l. 128-129; l. 541: The supplementary tables are not available for download and thus the screening for candidate mutations cannot be assessed. It remains unclear what else beyond “genes of most interest” (based on functional annotation and prediction of subcellular localization) appeared in the initial screen.

l. 161: The phrase ‘controlling root development’ is misleading. Obviously, the deletion of SSR1 does have a severe root phenotype but this does not imply a direct function of SSR1 in control of root growth. The phenotype very likely is just pleiotropic and no surprise if basic metabolic functions like energy supply are compromised. The term ‘control’ thus should be avoided.

l. 201; Fig. 2A: The presented data are not fully convincing because the fluorescence is rather weak and the labelling pattern for mitochondria does not seem typical in terms of organelle shape and number. It would be much better if a maximum projection of multiple individual sections was presented. nVenus and cCFP may well reconstitute, but the reconstituted protein is certainly not a GFP as indicated in the figure. It would be more appropriate to label this rather e.g. as ‘reconstituted FP’

l. 379: ‘ed’?

l. 384-386: Induction of SSR1 under multiple stress conditions may be more likely related to a broad chaperone function rather than to its apparent role in FeS assembly. This should be discussed.

l. 404, Fig. S7A: a counterstain for mitochondria is missing.

Reviewer #3: The manuscript describes the molecular function of SHORT AND SWOLLEN ROOT1 (SSR1) and its association with the mitochondrial iron-sulfur cluster machinery.

A genetic screen is used to identify suppressors of one of the mutations (ssr1-2) of SSR1. Two suppressors of the short growth phenotype, linked to the ssr1-2 mutation, are found, located in HSCA2 and ISU1. The association between SSR1 and HSCA2 and ISU1 is demonstrated in vivo and in vitro. In addition, the role of SSR1 as a chaperone is investigated. It is also shown that the 2 suppressors restore the mitochondrial defects previously identified in the ssr1 mutants.

The origin of SSR1 is discussed.

1) The interaction between SSR1 and the mitochondrial Fe-S machinery has never been described so far and is clearly supported by experimental data. However, some of them could be more convincing.

ex:

- the BiFC images are very weak. Is it linked to a weak expression of the two constructs? How many protoplasts were examined?

- the explanation about the BN-PAGE could be improved. What is the expected size of the HSCA2-SSR1-ISU1 complex? Is this size compatible with the 440 kDa signal or the one below 300 kDa? Is is made in E coli or in Arabidopsis? Could the association be improved by using different concentrations of the detergent (Tween-20) or the use of another detergent? Is it feasible to make MS on these spots on the first dimension of the gel to identify other components of the 440 kDa signal? What is the meaning of the number of hours on the top of the gels?

- would it be possible to make structural prediction about the interactions (alphafold) and what would be the scores of these interactions, using wild type and mutant versions of HSCA2 and ISU1?

2) the phenotype of the ssr1-1 and -2 mutants could be presented in the introduction. This would help to understand what is new or not about the physiological role of SSR1 described in the result section.

3) What is the definition of a 'novel' gene. TPR proteins is a large family. Could additional info could be brought?

Additional comments

- Could the authors comment about the semi-dominant inheritance of SU2?

- iron content: measurements against g DW or TOC would be more reliable. Are there other metals (S?, Zn) that show different contents?

- a supplemental file with the position of the suppressor mutations for both HSCA2 and ISU1 could be added.

- Line 520; incomplete unit

- transcriptomics data (Fig5A) should be deposited in a public repository.

- Fig6B: comment about the decrease of SDH2.

**Have all data underlying the figures and results presented in the manuscript been provided?**

Reviewer #1: Yes

Reviewer #2: **No: **supplementary tables were not accessible from the link in the manuscript file

Reviewer #3: **No: **The transcriptomics data of Fig 5A

PLOS authors have the option to publish the peer review history of their article (what does this mean?). If published, this will include your full peer review and any attached files.

Reviewer #1: No

Reviewer #2: No

Reviewer #3: No

---

## [Decision Letter · Decision Letter 1]

17 Dec 2024

PGENETICS-D-24-00725R1

The plant-specific cochaperone SSR1 affects root elongation by modulating the mitochondrial iron-sulfur cluster assembly machinery

PLOS Genetics

Dear Dr. Feng,

Thank you for submitting your manuscript to PLOS Genetics. After careful consideration, we feel that it has merit but does not fully meet PLOS Genetics's publication criteria as it currently stands. Therefore, we invite you to submit a revised version of the manuscript that addresses the points raised during the review process.

Please submit your revised manuscript within 30 days Jan 16 2025 11:59PM. If you will need more time than this to complete your revisions, please reply to this message or contact the journal office at plosgenetics@plos.org. Please include the following items when submitting your revised manuscript:

We look forward to receiving your revised manuscript.

Kind regards,

Nicolas Rouhier

Guest Editor

PLOS Genetics

Claudia Köhler

Section Editor

PLOS Genetics

Aimée Dudley

Editor-in-Chief

PLOS Genetics

Anne Goriely

Editor-in-Chief

PLOS Genetics

**Additional Editor Comments:**

Dear Dr Feng,

I agree with the reviewers that the content of the paper has been improved during this round of revisions, but that both some parts of the text and some figures would benefit from rigorous editing. All other points raised by the reviewers should be addressed.

**Journal Requirements:**

At this stage, the following Authors/Authors require contributions: Xuanjun Feng, Yue Hu, Tao Xie, Huiling Han, Diana Bonea, Lijuan Zeng, Jie Liu, Wenhan Ying, Bona Mu, Yuanyuan Cai, Min Zhang, Yanli Lu, Rongmin Zhao, and Xuejun Hua. Please ensure that the full contributions of each author are acknowledged in the "Add/Edit/Remove Authors" section of our submission form.

The list of CRediT author contributions may be found here: https://journals.plos.org/plosgenetics/s/authorship#loc-author-contributions

2) We have noticed that you have uploaded Supporting Information files, but you have not included a complete list of legends. Please include the legend for "Appendix Materials".

3) We notice that your supplementary figures are uploaded with the file type 'Figure'. Please amend the file type to 'Supporting Information'. Please ensure that each Supporting Information file has a legend listed in the manuscript after the references list.

4) We notice that your supplementary Figures are included in the manuscript file. Please remove them and upload them with the file type 'Supporting Information'. Please ensure that each Supporting Information file has a legend listed in the manuscript after the references list.

**Reviewers' comments:**

Reviewer's Responses to Questions

Reviewer #1: I have checked the response of the authors to my concerns. Here is my evaluation:

1. Improved, but still borderline quality of the pictures.

2. The authors provide a lengthy explanation for my concern. One may believe this or not. Apparently, the issue is more complex than appreciated.

3. The authors’ answer (“we did not indicate or claim that root tips absorb more iron.”) is incorrect: In two parts of the old and new manuscript, the authors mention the “iron accumulation in root tips”: Abstract, line 30, and lanes 325ff (“eight times more”, old manuscript) or lane 346f (“had higher iron content in the tips”, new manuscript). Why was this text changed and the quantitation removed? This point needs further careful reassessment.

4. I accept that not much is known about this topic. I did not get, however, if the authors have added any statement addressing this aspect to the manuscript.

5. Relationship between SSR1 and HSCB: Do the authors refer to lines 506-516 (the authors’ callout lines 460-479 are incorrect)? This part is a valuable addition, despite its vague statement. I am sure that the reader is still puzzled in what the role SSR1 is performing (chaperone or now co-chaperone?) and how its function compares to that of HSCB.

Comments:

1. Resolved (but the mentioned lines are wrong; also the lines mentioned in the following points; hence it is difficult to judge the responses).

2. Resolved (despite the wrong line callout), even though I would have preferred the mentioning of the major species studied.

3. Slightly improved only. Still incorrect parts: i) New evidence accumulated over the past decade does not confirm a role of FXN in iron transfer to ISCU. The statement “frataxin/Yfh1/cyaY are proposed to be iron donors” should either be deleted or indicated that this view was refuted by studies in the last decade. ii) The statement “transferred to recipient apo-proteins (such as glutaredoxin Grx5)” should be changed because Grx5 is not a recipient FeS protein (like aconitase). Rather, it is a cluster transfer protein. iii) In lanes 95f, a sentence from my initial review erroneously entered the manuscript (“The reader is left alone with the function and localization of NFS2 and ABA3 (both are known”.). Please remove, and explain the localization and function of NFS2 (plastid and SufS-like) and ABA3 (cytosol and MoCo synthesis).

4. Comment included in point 3.

5. Comment included in point 3.

6. Resolved.

7. There are still many cases where singular and plural “s” are wrong.

Overall, the authors have improved their manuscript. However, as mentioned above, there are essential parts that have to be carefully re-addressed before I can recommend publication.

Reviewer #2: The manuscript has been revised according to comments provided in the first round of reviewing. With these revisions some but not all shortcomings have been fixed.

You argue that the expression of auxin-related geens and the translational resonse of SSR1 to environmental stress is important to the story. I see you point that this links to your earlier publications on the characterization of the mutant. It is certainly a nice confirmation that the suppressor mutants also reactivate the expression of auxin-related genes. Yet, the effects are merely pleiotropic and don't help much to understand the function of SSR1, but rather distract the reader's attention from the core message.

The description of isolated suppressors and particularly the number of these repressors has revised. Yet, it is still confusing when in the abstract it is stated that 'eight kinds' suppressors have been isolated and in the first part of the Results section it is said that a dozen suppressors have been isolated (it is noted that in the rebuttal it is now still claimed ‘more than a dozen’). If eight of these 12 suppressors are designated sus1 to sus8, what is the designation of the remaining four suppressors? If these additional suppressors beyond sus1 and sus2 are considered important to be mentioned here, the respective identities should be revealed together with some reasoning for not following them further at this point. In the Abstract the reader is now referred to 'eight kinds of suppressors'. The phrase 'eight kinds of suppressors' suggest that these are different. Later, in the Results section, it turns out that seven out of these eight are mutants of either HSCA2 (2) or ISU1 (5), which I would count as two kinds of mutants. It is not obvious why you not simply focus on these seven mutants. You can still indicate that you have isolated other mutants but concentrate further detailed analysis on these seven mutants (or in fact one representative of each) because the large number of mutants in just two genes known to be involved in the same physiological process is highly intriguing.

Additional points:

It should be considered mentioning in the manuscript that attempts for structural modelling of SSR1, IU1 and HSCA2 including their interactions has been attempted but did not provide conclusive results.

According to your response letter, the two sentences in lines 94-96 with largely redundant information have been revised. This statement is not correct as the very same sentences are still in the text, now in lines 115-117.

A number of line numbers mentioned in the rebuttal do no match the respective line numbers in the manuscript text.

The brightness of the images in Fig. 2A have been adjusted for clarity. Such image modification procedures, even if done only for carity, should be indicated in the legend. The annotation 'FP' for the reconstituted protein is certainly more appropriate than 'GFP in the earlier text version. Yet, 'FP' should be explained in the legend.

l.48: To make the Abstract text consitent with the title and the Discussion, SSR1 should be called a cochaperone here.

l.95-96: Not clear to me what this sentence is meant to say. Delete or rephrase.

l.165-171: The text should be revised according to the comments provided above already; 'five additional types of mutants does not make sense because they apparently all are mutants of just two genes.

l.464: The exact number of Fe-S proteins is not known because there are still new proteins being dicovered as cluster coordinating. The total number, however, will almost certainly be much closer to 100 than 'hundreds' as stated here. The number in mitochondria, which are considered here, is much lower. Thus it would be sensible to tone this down a little.

l.509: 'We do not believe this is likely.' Delete this sentence here and discuss the data first. Draw a conclusion at the very end.

l.517: 'stress-adapting' should be 'stress adaptation'

Reviewer #3: The manuscript has been improved following revision by the authors.

However, there are still some points that could be clarified.

Line 95-96: to be completed.

Transcriptomics data. Providing a Table is fine but, in addition, raw data should be deposited on a public repository (NCBI, …). Raw data can be made accessible to the public only upon acceptance of the manuscript. An accession number should be provided in the manuscript.

Semi-dominance: I’m fine with the explanation. Fig S1 could be completed (panel D) in support to the explanation provided concerning the interaction between the WT and the mutated copies of ISU1 and HCSA2 in the presence and in the absence of SRR1.

Concerning the discussion, there would be 2 roles for SRR1: one described here in the ISC machinery, and one in the response to osmotic stress and proline treatment. How these two roles could be combined in a single protein ? Could they be linked to the structure of SRR1? TPR proteins may be composed of different domains (for example when they play a role in RNA edition). Is it the case for SRR1?

**Have all data underlying the figures and results presented in the manuscript been provided?**

Reviewer #1: Yes

Reviewer #2: Yes

Reviewer #3: **No: **public repository should be used for the raw data of the RNA seq

PLOS authors have the option to publish the peer review history of their article (what does this mean?). If published, this will include your full peer review and any attached files.

Reviewer #1: No

Reviewer #2: No

Reviewer #3: No

**Figure resubmission:**
---

## [Editor Report · Decision Letter 2]

22 Jan 2025

PGENETICS-D-24-00725R2

Plant-specific cochaperone SSR1 affects root elongation by modulating the mitochondrial iron-sulfur cluster assembly machinery

PLOS Genetics

Dear Dr. Feng,

Thank you for submitting your manuscript to PLOS Genetics. After careful consideration, we feel that it has merit but does not fully meet PLOS Genetics's publication criteria as it currently stands. Therefore, we invite you to submit a revised version of the manuscript that addresses the points raised during the review process.

Please submit your revised manuscript within 60 days. If you will need more time than this to complete your revisions, please reply to this message or contact the journal office at plosgenetics@plos.org. Please include the following items when submitting your revised manuscript:

We look forward to receiving your revised manuscript.

Kind regards,

Nicolas Rouhier

Guest Editor

PLOS Genetics

Tanja Slotte

Section Editor

PLOS Genetics

Aimée Dudley

Editor-in-Chief

PLOS Genetics

Anne Goriely

Editor-in-Chief

PLOS Genetics

**Additional Editor Comments :**

Dear Dr Feng,

I have carefully checked your last responses to the reviewer’s comments and noticed that you are unable to provide the raw data for the transcriptomic analysis. This is not in line with the PLOS data availability policy and more generally with the standards of data sharing and reproducibility in the field (see the MIAME (Minimum Information About a Microarray Experiment) and MINSEQE (Minimum Information About a Next-generation Sequencing Experiment) recommendations). As I believe that the paper remains of sufficient quality without these transcriptomic data, I suggest that you remove these results from the paper and submit a modified version.

You have the option of declining this proposal, but in this case the article cannot be published in PLos genetics.

Best Regards

Nicolas Rouhier

PLOS journals require authors to make all data necessary to replicate their study’s findings publicly available without restriction at the time of publication. In line with the Associate Editor's comments, we therefore ask that you modify the manuscript in order to meet these requirements. 

**Figure resubmission:**
---

## [Editor Report · Decision Letter 3]

29 Jan 2025

Dear Dr Feng,

We are pleased to inform you that your manuscript entitled "Plant-specific cochaperone SSR1 affects root elongation by modulating the mitochondrial iron-sulfur cluster assembly machinery" has been editorially accepted for publication in PLOS Genetics. Congratulations!

Yours sincerely,

Nicolas Rouhier

Guest Editor

PLOS Genetics

Tanja Slotte

Section Editor

PLOS Genetics

Aimée Dudley

Editor-in-Chief

PLOS Genetics

Anne Goriely

Editor-in-Chief

PLOS Genetics

Comments from the reviewers (if applicable):

**Data Deposition**

http://datadryad.org/submit?journalID=pgenetics&manu=PGENETICS-D-24-00725R3

**Press Queries**

---

## [Editor Report · Acceptance letter]

31 Jan 2025

PGENETICS-D-24-00725R3 

Plant-specific cochaperone SSR1 affects root elongation by modulating the mitochondrial iron-sulfur cluster assembly machinery 

Dear Dr Feng, 

We are pleased to inform you that your manuscript entitled "Plant-specific cochaperone SSR1 affects root elongation by modulating the mitochondrial iron-sulfur cluster assembly machinery" has been formally accepted for publication in PLOS Genetics! Your manuscript is now with our production department and you will be notified of the publication date in due course.

With kind regards,

Anita Estes

PLOS Genetics

On behalf of:
